# GlobeDiff: State Diffusion Process for Partial Observability in Multi-Agent Systems

**Yiqin Yang**[1]*, **Xu Yang**[2]*, **Yuhua Jiang**[2], **Ni Mu**[2], **Hao Hu**[3], **Runpeng Xie**[1], **Ziyou Zhang**[2],
**Siyuan Li**[5], **Yuan-Hua Ni**[4], **Qianchuan Zhao**[2]†, **Bo Xu**[1]†

[1]The Key Laboratory of Cognition and Decision Intelligence for Complex Systems,
  Institute of Automation, Chinese Academy of Sciences
[2]Tsinghua University    [3]Moonshot AI    [4]Nankai University
[5]Faculty of Computing, Harbin Institute of Technology

## Abstract

In the realm of multi-agent systems, the challenge of *partial observability* is a critical barrier to effective coordination and decision-making. Existing approaches, such as belief state estimation and inter-agent communication, often fall short. Belief-based methods are limited by their focus on past experiences without fully leveraging global information, while communication methods often lack a robust model to effectively utilize the auxiliary information they provide. To solve this issue, we propose Global State Diffusion Algorithm (GlobeDiff) to infer the global state based on the local observations. By formulating the state inference process as a multi-modal diffusion process, GlobeDiff overcomes ambiguities in state estimation while simultaneously inferring the global state with high fidelity. We prove that the estimation error of GlobeDiff under both unimodal and multi-modal distributions can be bounded. Extensive experimental results demonstrate that GlobeDiff achieves superior performance and is capable of accurately inferring the global state.

## 1 Introduction

Multi-Agent Reinforcement Learning (MARL) has driven significant progress in complex domains like robotics (Wang et al., 2022; Lee et al., 2022) and autonomous systems (Zhang et al., 2024; Zhou et al., 2022), enabling agents to learn sophisticated collaborative policies (Feng et al., 2024; Wang et al., 2024b). However, a fundamental and persistent barrier to effective multi-agent coordination is the problem of partial observability (PO), where each agent's view is limited (Amato et al., 2013; Omidshafiei et al., 2017; Srinivasan et al., 2018), and the true global state of the system is unknown. This challenge, formally captured in the Decentralized Partially Observable Markov Decision Process (Dec-POMDP) framework (Oliehoek et al., 2016), forces agents to act under uncertainty, often leading to suboptimal or conflicting decisions (Spaan, 2012).

The core difficulty of partial observability lies in the profound ambiguity it creates: a single agent's local observation can be consistent with numerous, often dramatically different, global states. This creates a challenging one-to-many mapping problem for state inference. Existing approaches have attempted to resolve this ambiguity using discriminative models, such as recurrent networks or Transformers (Hausknecht & Stone, 2015; Kapturowski et al., 2018), which learn to predict a single, most likely global state from a history of local observations. However, this approach is fundamentally flawed. By collapsing a rich distribution of possibilities into a single point estimate, these methods suffer from mode collapse. They either average distinct plausible states into a single, nonsensical representation or arbitrarily commit to one possibility while ignoring others, failing to capture the true uncertainty of the environment.

The central thesis of this paper is that the one-to-many ambiguity inherent in global state inference is best addressed not by discriminative prediction, but by generative modeling (Goodfellow et al.,

---

*These authors contributed equally.
†Corresponding author.

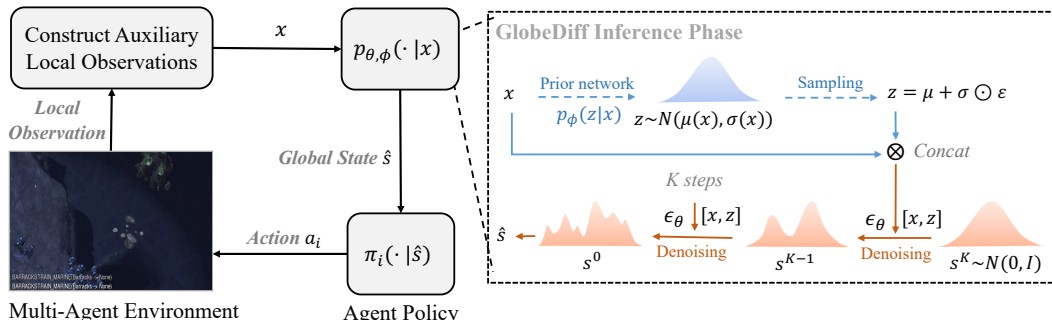

Figure 1: **The overall framework of GlobeDiff.** During the execution phase, we first construct auxiliary local observations $x$ and then infer the global state $\hat{s}$ using GlobeDiff. Agents make decisions based on the inferred global state $\hat{s}$.

2014; Ho et al., 2020; Song et al., 2020). Instead of forcing a collapse to a single mode, a generative approach can learn the entire conditional distribution of plausible global states. This allows an agent to reason over the full spectrum of possibilities, a capability that is critical for robust decision-making under uncertainty. By sampling from this learned distribution, our method can generate high-fidelity hypotheses about the global state, directly confronting the multi-modality of the problem.

To realize this vision, we introduce the Global State Diffusion Algorithm (GlobeDiff), a novel framework that operationalizes this generative insight using a conditional diffusion model. GlobeDiff formulates global state inference as a denoising process, learning to reverse a diffusion process that gradually corrupts the global state into noise. Conditioned on an agent's local information (such as its own observations or communicated messages (Kim et al., 2019; Jiang & Lu, 2018)), GlobeDiff can generate a diverse and realistic set of potential global states. This approach not only provides a more principled solution to the one-to-many mapping problem but also integrates seamlessly into existing MARL frameworks like Centralized Training with Decentralized Execution (CTDE). Our contributions are:

- We identify and frame the core challenge of partial observability as a one-to-many mapping problem, highlighting the limitations of existing discriminative approaches.
- We propose GlobeDiff, the framework to leverage conditional diffusion models for generative global state inference in MARL, offering a robust solution to mode collapse.
- We empirically demonstrate that GlobeDiff significantly outperforms state-of-the-art baselines on challenging multi-agent benchmarks, validating the power of the generative approach.

## 2    RELATED WORK

**Partial Observability**    To solve the PO problem, particularly in Dec-POMDPs, existing research can be divided into two categories: belief state estimation and explicit communication. First, to model uncertainty in multi-agent systems, the concept of belief state has been introduced to estimate the state of the environment or other agents (MacDermed & Isbell, 2013; Muglich et al., 2022; Varakantham et al., 2006). For example, given the effectiveness in handling temporal sequences, RNNs are used to integrate local observation histories over time, providing agents with long-term memory (Hausknecht & Stone, 2015; Kapturowski et al., 2018; Wen et al., 2022). However, estimation errors accumulate over time, leading to insufficient information in complex systems and hindering a comprehensive understanding of the global state. In contrast to inferring or estimating the global belief state, inter-agent communication has been introduced to directly acquire information from other agents and expand the receptive field of individual agents (Das et al., 2019; Singh et al., 2018; Zhang et al., 2019; Kim et al., 2019; Jiang & Lu, 2018). However, these approaches suffer from high communication costs and complex protocol design.

**Diffusion Model for RL**    Diffusion models leverage a denoising framework and effectively reverse multi-step noise processes to generate new data (Ho et al., 2020; Song et al., 2020). These models

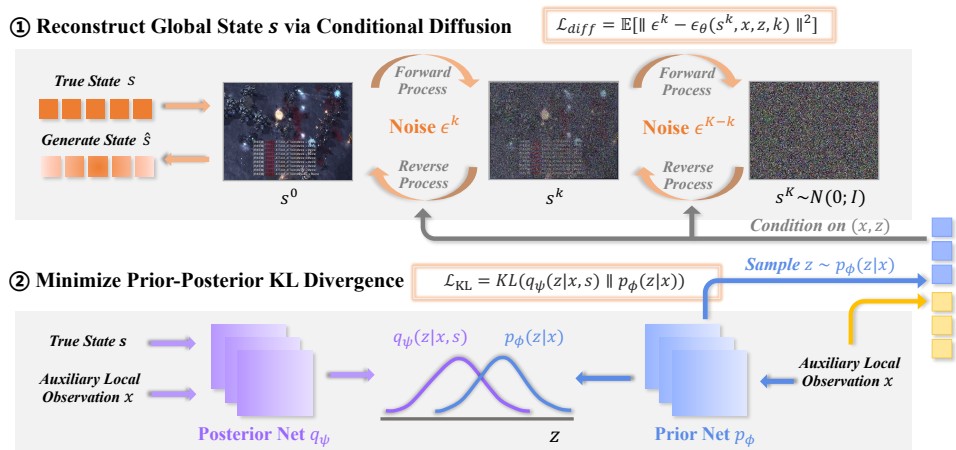

Figure 2: The training process of Globediff is divided into two parts: minimizing the difference between the prior network $p_\phi$ and the posterior network $q_\psi$, and then training the diffusion model based on the forward and backward process.

have increasingly been integrated into sequential decision-making tasks to improve performance and sample efficiency, particularly in single-agent and offline reinforcement learning. For example, diffusion models have been employed as planners, encoding dynamic environmental information and generating multi-step optimal trajectories (Janner et al., 2022; Liang et al., 2023; He et al., 2023; Ajay et al.; Chi et al., 2023). This integration helps mitigate compound errors in auto-regressive sequence planning and facilitates better long-term decision-making. Additionally, diffusion models have recently been applied to address the PO problem in multi-agent systems (Wang et al., 2024a). However, they focuses on approximating belief distributions via shared attractors, but do not explicitly model the intrinsic one-to-many mapping from local observations to the global state.

## 3  PRELIMINARIES

**Dec-POMDPs**  We consider Dec-POMDP as a standard model consisting of a tuple $\mathcal{G} =<\mathcal{S}, \mathcal{A}, \mathcal{P}, \mathcal{R}, \mathcal{U}, \mathcal{O}, \gamma >$ for cooperative multi-agent tasks. Within $\mathcal{G}$, $s \in \mathcal{S}$ denotes the global state of the environment. Each agent $i \in N := 1, ..., n$ chooses an action $a_i \in \mathcal{A}$ at each time, forming a joint action $\boldsymbol{a} \in \mathcal{A}^n$. The state transition function $\mathcal{P}(s'|s, \boldsymbol{a}) : \mathcal{S} \times \mathcal{A}^n \times \mathcal{S} \to [0, 1]$ gives a transition to the environment. The reward function $\mathcal{R}(s, \boldsymbol{a}) : \mathcal{S} \times \mathcal{A}^n \to \mathbb{R}$ is shared among all agents and $\gamma \in [0, 1)$ is the discount factor.

In a partially observable scenario, each agent has individual observations $o \in \mathcal{O}$ according to the observation function $\mathcal{U}(s, a) : \mathcal{S} \times \mathcal{A} \to \mathcal{O}$. Each agent makes decision based on a stochastic policy $\pi_{\vartheta_i}(a_i \mid o_i)$ parameterized by $\vartheta_i$: $\mathcal{O} \times \mathcal{A} \to [0, 1]$. The joint value function can be defined as $Q_{tot}^{\boldsymbol{\pi}}(s_t, \boldsymbol{a}_t) = \mathbb{E}_{s_{t+1}:\infty, \boldsymbol{a}_{t+1}:\infty}[\sum_{i=0}^{\infty} \gamma^i r_{t+i}|s_t = s, \boldsymbol{a}_t = \boldsymbol{a}, \boldsymbol{\pi}]$, where $\boldsymbol{\pi}$ is a joint policy with parameters $\vartheta =< \vartheta_1, ..., \vartheta_n >$.

**Generative Model for Global State Inference**  We aim to learn a mapping from the auxiliary local observations $x$ to the global state $s$ based on the generative model $p_\theta(s \mid x)$, where auxiliary local observations $x$ are composed of local observations $o$. Therefore, agents can make decisions based on the global state $s$ rather than the local observations $o$ during execution by leveraging the generative model $p_\theta$, thereby overcoming the limitations of partial observability. In this work, we consider two scenarios for generating auxiliary states. First, if local observations $o$ are information-rich, we can infer the global state based on the individual agent's historical trajectory, in which case the auxiliary local observations $x_t$ is formulated as the integration of observations $o_t^i$ over the past $m$ steps:

$$x_t = \{o_{t-m}^i, \quad o_{t-m+1}^i, \quad \cdots, o_t^i\}, \tag{1}$$

where $o_t^i$ denotes the local observation of agent $i$ at time step $t$. On the other hand, if local observations provide limited information, it becomes challenging to infer the global state based on the individual

agent's historical trajectory. In such scenarios, we enable communication between agents, and consequently, the auxiliary local observation $x_t$ is constructed from their joint observations:

$$x_t = \{o_t^1, \quad o_t^2, \quad \cdots, o_t^n\}. \tag{2}$$

## 4 METHOD

Our methodological approach is designed to tackle the fundamental ambiguity of partial observability: a single local observation $x$ can correspond to many different, yet plausible, global states $s$. A naive conditional generative model $p(s|x)$ would struggle with this one-to-many mapping, likely averaging over the possibilities and producing a blurry global state. To address this, we introduce a key architectural choice: a latent variable $z$. The intuition is to use $z$ as a mode selector. Instead of asking the model to solve the ambiguous problem of generating $s$ from $x$, we ask it to solve the well-posed problem of generating $s$ from both $x$ and $z$. The latent variable $z$ provides the specific context needed to select one particular plausible global state from the distribution of possibilities. This transforms the problem into learning a conditional diffusion model $p(s|x,z)$.

This design, however, introduces a new challenge: how do we obtain a meaningful $z$ during inference when we only have the local observation $x$? We solve this by bridging the gap between training and inference. During training, we have access to the ground-truth global state $s$, which allows us to train an posterior network, $q(z|x,s)$, that learns the ideal $z$ required to reconstruct $s$ from $x$. For inference, we train a separate prior network, $p(z|x)$, which predicts a useful $z$ using only $x$. The following sections will detail the mathematical formulation of this diffusion process.

### 4.1 GLOBAL STATE DIFFUSION PROCESS

#### 4.1.1 TRAINING

Inspired by nonequilibrium thermodynamics (Sohl-Dickstein et al., 2015), we attempt to formulate $p_\theta(s \mid x)$ as a diffusion process. However, since the observation function $\mathcal{U}$ does not assume the unique mapping between elements of the input set $(\mathcal{S} \times \mathcal{A})$ and output set $\mathcal{O}$, different global states may be mapped to the same local observation. Therefore, when inferring global states from local observations, the ambiguity issue caused by the non-unique mapping significantly decreases the accuracy of global state inference. To address this issue, we introduce a latent variable $z$ that allows the diffusion process to map single input $x$ to multiple outputs $s$, that is modeling a one-to-many conditional generative distribution $p_{\theta,\phi}(s \mid x)$:

$$p_{\theta,\phi}(s \mid x) = \int p_\theta(s \mid x, z)p_\phi(z \mid x)dz, \tag{3}$$

where $p_\phi(z \mid x)$ is a conditional prior. Then, we introduce an approximate posterior $q_\psi(z \mid x, s)$ and derive the following equation based on Jensen's inequality (Kingma & Welling, 2014):

$$
\begin{aligned}
\log p_{\theta,\phi}(s \mid x) &= \log \int p_\theta(s \mid x, z)p_\phi(z \mid x)dz \\
&= \log \int q_\psi(z \mid x, s)\frac{p_\theta(s \mid x, z)p_\phi(z \mid x)}{q_\psi(z \mid x, s)}dz \\
&\geq \mathbb{E}_{q_\psi}\left[\log \frac{p_\theta(s \mid x, z)p_\phi(z \mid x)}{q_\psi(z \mid x, s)}\right] \\
&= \mathbb{E}_{q_\psi}[\log p_\theta(s \mid x, z)] - \mathrm{KL}(q_\psi(z \mid x, s)\|p_\phi(z \mid x)).
\end{aligned}
\tag{4}
$$

For $p_\theta(s \mid x, z)$, in the forward process, we first sequentially introduce Gaussian noise $\epsilon$ to the global state $s$ according to the predefined variance:

$$q(s^k \mid s^{k-1}) = \mathcal{N}(s^k; \sqrt{1 - \beta^k}s^{k-1}, \beta^k\mathbf{I}), \tag{5}$$

where $k \in \{0, ..., K\}$ is the diffusion timestep, $\beta^k$ is the variance parameter, $s^0$ is the original state and $s^k$ is the state corrupted with $k$-step noise. For any $s^k$, we compute it from the original state $s^0$ without intermediate steps:

$$s^k = \sqrt{\overline{\alpha}^k}s^0 + \sqrt{1 - \overline{\alpha}^k}\epsilon(s^k, k), \tag{6}$$

where $\epsilon(s^k, k) \sim \mathcal{N}(\mathbf{0}, \mathbf{I})$ is the $k$-th step noise of the forward process and $\overline{\alpha}^k = \Pi_{i=1}^k \alpha^i$ with $\alpha^k = 1 - \beta^k$. Then, we represent the global state reference via the reverse process of the diffusion model as

$$p_\theta(s \mid x, z) = p_\theta(s^{0:K} \mid x, z) = \mathcal{N}(s^K; \mathbf{0}, \mathbf{I})\Pi_{k=1}^K p_\theta(s^{k-1} \mid s^k, x, z), \tag{7}$$

where the end sample of the reverse chain $s^0$ is the restored global state. Generally, $p_\theta(s^{k-1} \mid s^k, x, z)$ could be modeled as a Gaussian distribution $\mathcal{N}(s^{k-1}; \mu_\theta(s^k, x, z, k), \sigma_\theta(s^k, x, z, k))$. We follow Ho et al. (2020) to parameterize $p_\theta(s^{k-1} \mid s^k, x, z)$ as a noise prediction model with the covariance matrix fixed as $\sigma_\theta(s^k, x, z, k) = \beta^k \mathbf{I}$ and mean constructed as

$$\mu_\theta(s^k, x, z, k) = \frac{1}{\sqrt{\alpha^k}} \left( s^k - \frac{\beta^k}{\sqrt{1 - \overline{\alpha}^k}} \epsilon_\theta(s^k, x, z, k) \right). \tag{8}$$

We first sample $s^K \sim \mathcal{N}(\mathbf{0}, \mathbf{I})$ and then form the reverse diffusion chain parameterized by $\theta$ as

$$s^{k-1} \mid s^k = \frac{1}{\sqrt{\alpha^k}} \left( s^k - \frac{\beta^k}{\sqrt{1 - \overline{\alpha}^k}} \epsilon_\theta(s^k, x, z, k) \right) + \sqrt{\beta^k}\epsilon, \quad \epsilon \sim \mathcal{N}(\mathbf{0}, \mathbf{I}). \tag{9}$$

Therefore, the global state diffusion process is trained by minimizing the following loss function:

$$\mathcal{L}(\theta, \phi, \psi) = \mathbb{E}_{k \sim \mathcal{U}, \epsilon \sim \mathcal{N}(\mathbf{0}, \mathbf{I}), (s,x) \sim \mathcal{D}, z \sim q_\psi} \left[ \| \epsilon - \epsilon_\theta \left( \sqrt{\overline{\alpha}^k}s + \sqrt{1 - \overline{\alpha}^k}\epsilon, x, z, k \right) \|^2 \right] + \tag{10}$$
$$\beta_{\text{KL}}\text{KL}(q_\psi(z \mid x, s) \| p_\phi(z \mid x)),$$

where $\mathcal{U}$ is a uniform distribution over the discrete set as $\{1, ..., K\}$, $\mathcal{D}$ denotes the datasets and $\beta_{\text{KL}}$ is a hyperparameter. The overall training process of GlobeDiff is shown in Figure 2.

### 4.1.2 INFERENCE

In the inference phase, each agent first obtains the latent variable $z$ via the encoder $p_\phi(z \mid x)$. Then, each agent initializes $s^K \sim \mathcal{N}(\mathbf{0}, \mathbf{I})$ and performs $K$ iterative denoising sampling steps. Note that $s^K$ is initialized as Gaussian noise, and $s^0$ obtained after $K$ denoising steps is the inferred global state. Crucially, no global information is utilized throughout the inference process. Consistent with the reverse process in Equation 9, the inference at the $k$-step is performed as follows:

$$s^{k-1} = \frac{1}{\sqrt{\alpha^k}} \left( s^k - \frac{\beta^k}{\sqrt{1 - \overline{\alpha}^k}} \epsilon_\theta(s^k, x, z, k) \right) + \sqrt{\beta^k}\epsilon, \tag{11}$$

where $\epsilon \sim \mathcal{N}(\mathbf{0}, \mathbf{I})$ represents standard Gaussian noise. After $K$ inference steps, $s^0$ becomes the inferred global state, where each agent makes decisions by $a_i = \pi_{\vartheta_i}(\cdot \mid s^0)$.

The aforementioned global state diffusion process exhibits the following characteristics. First, the diffusion process does not explicitly model the distribution of generated samples but implicitly learns it through the denoising network $\epsilon_\theta$. Therefore, the marginal of the reverse diffusion chain provides an expressive distribution that can capture complex distribution properties. Second, the proposed global state diffusion process can model the non-unique mapping relationship between local observations and the global state. Finally, the global state inference is conditioned on the auxiliary state, enabling sampling those global states relevant to local observations.

### 4.2 THEORETICAL ANALYSIS

Let $\hat{s}$ denote the global state generated from GlobeDiff. When the observation function $\mathcal{U}$ is injective, the mapping between $s$ and $x$ is one-to-one. Assuming that the denoising network $\epsilon_\theta$ and prior network $p_\phi$ are well trained, we prove that the estimation error of GlobeDiff can be bounded:

**Theorem 1** (Single-Sample Expectation Error Bound with Latent Variable). *Assume the trained model satisfies the following two assumptions. (1) Diffusion noise prediction MSE:* $\mathbb{E}_{s^k, x, z, k}[\|\epsilon_\theta(s^k, x, z, k) - \epsilon\|^2] \leq \delta^2$, *(2) Prior alignment:* $D_{KL}(p_\phi(z \mid x) \| p(z \mid x)) \leq \varepsilon_{KL}$. *Then, for any generated sample* $\hat{s} \sim p_{\theta,\phi}(s \mid x) = \int p_\theta(s \mid x, z)p_\phi(z \mid x)dz$ *and true sample* $s \sim p(s \mid x)$, *the expected squared error is bounded by:*

$$\mathbb{E} \left[ \|\hat{s} - s\|^2 \right] \leq 2W_2^2(p_{\theta,\phi}(s \mid x), p(s \mid x)) + 4Var(s \mid x), \tag{12}$$

where $W_2$ is the 2-Wasserstein distance between $p_{\theta,\phi}(s \mid x)$ and $p(s \mid x)$, $Var(s \mid x) = \mathbb{E}_{p(s|x)}\left[\|s - \mu_{s|x}\|^2\right]$ is the conditional variance and $\mu_{s|x} = \mathbb{E}_{p(s|x)}[s]$ is the conditional mean.

*Proof.* Please refer to Appendix B.1 for the detailed proof. $\square$

However, in practical scenarios, the mapping from $x$ to $s$ is typically one-to-many. We model this situation using a multi-modal Gaussian distribution, as presented in Theorem 2, and we prove that the error between the estimated state and the centers of the multi-modal distribution also admits a bounded error.

**Theorem 2** (Multi-Modal Error Bound with Latent Variable). *Under the following conditions: (1) The true conditional distribution $p(s \mid x) = \sum_{i=1}^{N} w_i \mathcal{N}(s; \mu_i(x), \Sigma_i(x))$ has $N$ modes with minimum inter-mode distance $D = \min_{i \neq j} \|\mu_i(x) - \mu_j(x)\| \geq 2\sqrt{d}$. (2) Mode separation condition: $D > 4\sqrt{C_1 K \delta^2 + C_2 \varepsilon_{KL} + \max_i Tr(\Sigma_i(x))}$ (3) The model satisfies $\mathbb{E}[\|\epsilon_\theta - \epsilon\|^2] \leq \delta^2$ and $D_{KL}(p_\phi(z \mid x)\|p(z \mid x)) \leq \varepsilon_{KL}$. Then, for any generated sample $\hat{s} \sim p_{\theta,\phi}(s \mid x)$, there exists a mode $\mu_j(x)$ such that:*

$$\mathbb{E}\left[\|\hat{s} - \mu_j(x)\|^2\right] \leq C_1 K \delta^2 + C_2 \varepsilon_{KL} + 2\max_i Tr(\Sigma_i(x)) + \mathcal{O}\left(e^{-D^2/(8\sigma_{max}^2)}\right), \quad (13)$$

*where $\sigma_{max}^2 = \max_i Tr(\Sigma_i(x))$, and $C_1, C_2$ are constants depending on the diffusion scheduler and latent space geometry.*

*Proof.* Please refer to Appendix B.3 for the detailed proof. $\square$

In essence, Theorem 1 offers a universal error bound, while Theorem 2 provides a stronger, more specialized guarantee for the multi-modal settings our method is designed for. These results provides strong theoretical support for our approach. Consistent with our theoretical findings, the empirical results depicted in Figure 5 quantitatively verify the high fidelity of our method in state reconstruction.

## 4.3 PRACTICAL IMPLEMENTATION

**Architecture** We adopt a model consisting of repeated convolutional residual blocks to implement the global state diffusion process. The overall architecture resembles the types of U-Nets (Ronneberger et al., 2015), but with two-dimensional spatial convolutions replaced by one-dimensional temporal convolutions. Because the model is fully convolutional, the horizon of the inference is determined not by the model architecture but by the input dimensionality. This model can change dynamically during inference if desired.

**Training Mechanism** In the practical implementation, we first train an initial global state diffusion model based on an offline dataset. Subsequently, during online execution, we continuously update the global state diffusion model with the collected data to compensate for the distribution mismatch between offline and online settings. This approach ensures that the global state diffusion model plays a role from the early stages of algorithm training, reducing the instability of MARL algorithms caused by the generative model. In addition, during integration with the CTDE mechanism, we employ the true global state in the policy training phase to reduce computational cost. During the decentralized execution phase, each agent makes decisions based on the inferred global state. The overall process is shown in Algorithm 1.

## 5 EXPERIMENTS

We designed our experiments to answer the following questions: *Q1:* Can our method accurately infer the global state from the local observations? *Q2:* Can the global states generated by our method improve the performance of the MARL algorithm? *Q3:* Does our method outperform other generative models?

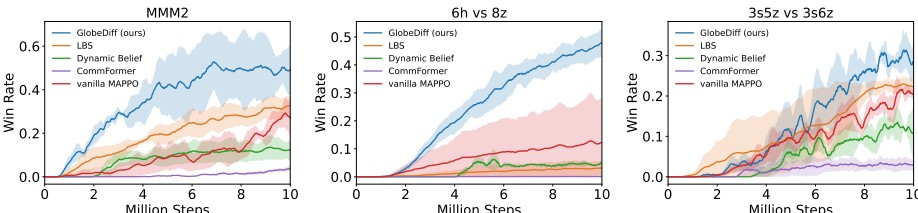

Figure 3: Comparison results with global state inference baselines in SMAC-v1 (PO) tasks with win rate over three random seeds.

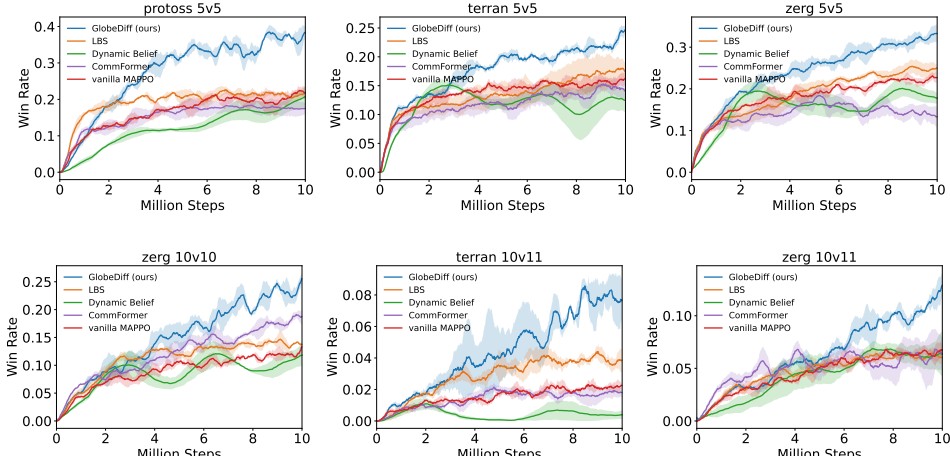

Figure 4: Comparison results with global state inference baselines in SMAC-v2 (PO) tasks with win rate over three random seeds.

## 5.1 SETUP

To answer these questions, we evaluate our method and baselines on SMAC, which is a cooperative MARL environment based on the real-time strategy game StarCraft II (Rashid et al., 2020). It includes various unique scenarios where agents can obtain local observations within a certain visual radius. The objective across all scenarios is to command allied units to eliminate enemy units. In addition, SMAC-v2 (Ellis et al.) further reduces the correlation between the local observation and the global state by adding random team compositions and random start positions.

**Benchmark for Partial Observability** Nevertheless, we are surprised to find that *the original SMAC environment is not well-suited for studying partial observability problems*. Specifically, we adjust the sight range of each agent from 9 to 3 in SMAC-v2 and run the standard MARL algorithm MAPPO. As shown in the Figure 8, the MAPPO's performance exhibits only a marginal decline (merely 0.03 drop) as the observation range narrows. This is because the local observations retain sufficient environmental information. Therefore, we modify the SMAC environment by removing enemy unit types and hit points from local observations to better evaluate partial observability issues, with this adapted environment named as SMAC-v1 (PO) and SMAC-v2 (PO).

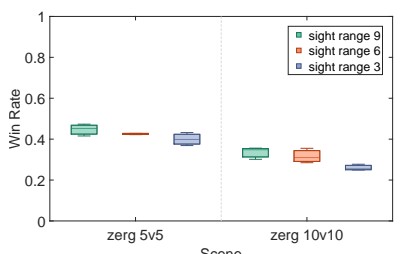

Figure 8: Win rate with various sight range in original SMAC tasks.

To ensure fair comparison, all experiments are conducted under identical environmental settings, with three random seeds employed per experiment.

**Baselines** We compare our approach with three representative baselines. Learned Belief Search (LBS) (Hu et al., 2021) learns an auto-regressive counterfactual belief model to approximate hidden information given the trajectory of an agent and uses a public-private policy architecture with RNNs to

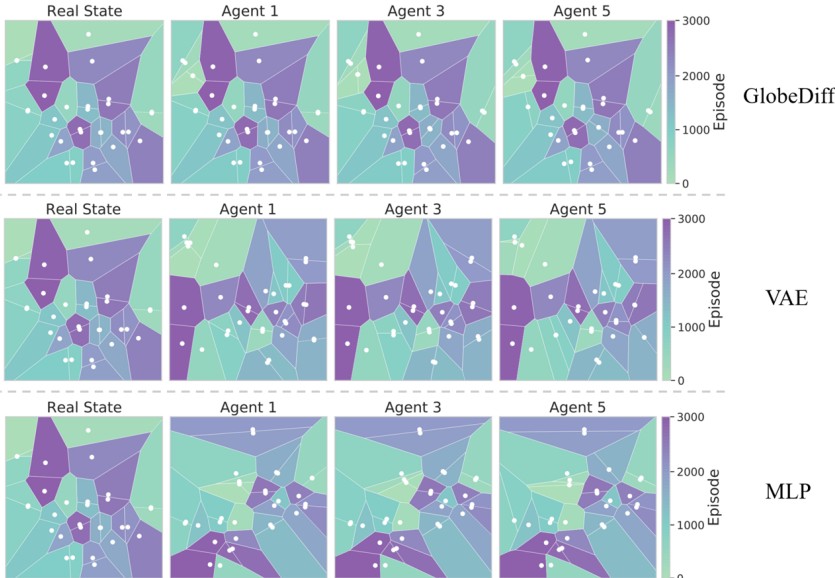

Figure 5: **Visualization of global states generated by GlobeDiff, VAE and MLP.** The first plot displays true states and subsequent plots show inferred states per agent. White points denote individual states with polygons highlighting local neighborhoods. Gradient shading (light green to purple) indicates training progression. The similarity between the polygon structures of the inferred and true states reflects the predicted quality.

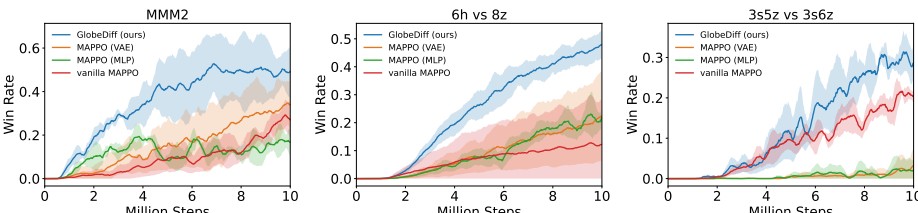

Figure 6: Comparison results with generative model baselines in SMAC-v1 (PO) tasks with win rate over three random seeds.

encode action-observation histories. Dynamic belief (Zhai et al., 2023) predicts the evolving policies of other agents from recent action-observation histories using a variational inference framework. CommFormer (Hu et al., 2024) learns a dynamic communication graph via continuous relaxation and attention-based message passing, jointly optimizing the graph and policy parameters. In the practical implementation, we combine our method and baselines with the standard MARL algorithm MAPPO. Please refer to Appendix D for the detailed implementation setting.

## 5.2 MAIN RESULTS

**Answer for Question 1:** To intuitively demonstrate the GlobeDiff's ability to infer the global state, we visualize both the true global states and the states reconstructed by GlobeDiff in the SMAC-V2 (PO) Zerg 5v5 scenario. For trajectories sampled during online training, we apply t-SNE (Maaten & Hinton, 2008) to project their high-dimensional states into a two-dimensional space. In Figure 5, the first plot shows the ground-truth states, while the subsequent plots depict the global states inferred by each agent. Each white point represents a specific state. To facilitate a visual comparison of the underlying structure, we overlay Voronoi polygons (Balzer et al., 2005). Each polygon defines the region of space closest to a single state point, effectively highlighting the local neighborhood. This allows for an intuitive assessment of accuracy: the more the polygon shapes in an inferred plot resemble those in the ground-truth plot, the better the state representation. The background is shaded with a gradient from light green to deep purple, where darker regions indicate episodes that occurred later in the online training process.

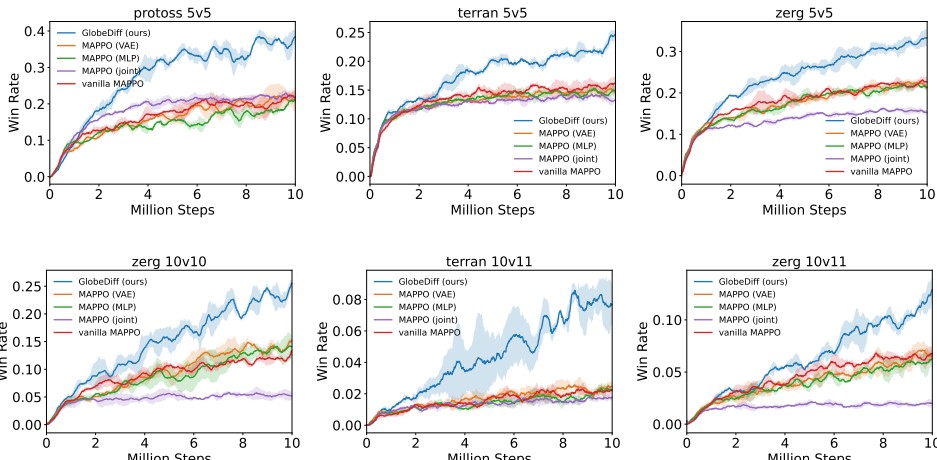

Figure 7: Comparison results with generative model baselines in SMAC-v2 (PO) tasks with win rate over three random seeds.

The similarity between the Voronoi polygon structures of the inferred and true states reflects the reconstruction quality. As shown in the Figure 5, the inferred states closely match the real states, indicating that GlobeDiff effectively enables agents to infer the global state from local information. Moreover, as training progresses (i.e., as the background color deepens), the inferred polygons become increasingly similar to the true ones, demonstrating steady improvement in reconstruction performance.

**Answer for Question 2:** We conduct experiments on SMAC-v1 (PO) and SMAC-v2 (PO) respectively. Specifically, for SMAC-v1 (PO), we derive auxiliary information based on the individual agent's historical trajectory, as detailed in Equation 1. For SMAC-v2 (PO), we leverage the communication between agents to construct auxiliary information, as outlined in Equation 2. The experimental results in Figure 3 and Figure 4 show that GlobeDiff consistently and significantly outperforms baseline algorithms in most maps. This performance gap can be attributed to the constrained capacity of baseline algorithms in modeling complex multi-modal distributions. For example, LBS tends to gradually accumulate errors when inferring belief states in long-horizon tasks. The inference process of Dynamic Belief remains unimodal, restricting its ability to capture the multi-modal global state distributions. CommFormer requires explicit communication and accurate message aggregation, which can be unreliable under severe partial observability. In contrast, GlobeDiff formulates the state inference process as a multi-model denoising procedure, implicitly modeling complex distributions within the noise network. This offers a highly expressive model that enhances algorithmic performance through accurate global state inference.

**Answer for Question 3:** Same with the scenario described in **Answer for Question 2**, we conduct experiments on SMAC-v1 (PO) and SMAC-v2 (PO) respectively. We employ VAE (Kingma & Welling, 2014) and MLP as comparative baselines for generative models. Specifically, we replace the GlobeDiff with the conditional VAE or an MLP, while keeping everything else unchanged, and name them respectively as MAPPO (VAE) and MAPPO (MLP). Furthermore, in the SMAC-v2 (PO) scenario, where agents can obtain information from adjacent agents, we incorporate the agents' joint observations as policy input. This approach is named as MAPPO (Joint) and serves as an additional base-

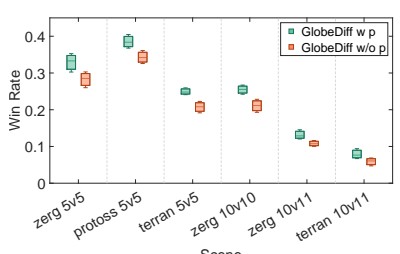

Figure 9: Ablation of prior network.

line to evaluate the role of generative models in state inference. Please refer to Appendix D for detailed description.

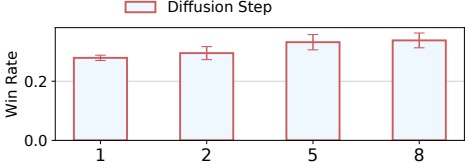 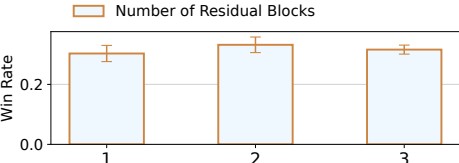

Figure 10: Ablation study for diffusion step and residual blocks on the zerg 5v5 task.

The experimental results in Figure 6 and Figure 7 show that GlobeDiff outperforms all baselines on the super-hard maps. The MLP and VAE show no significant performance improvement over vanilla MAPPO in most maps, which is attributed to their limited representation capacity. Moreover, in SMAC-v2 (PO), MAPPO (Joint) performs worse than vanilla MAPPO in some maps. This shows the necessity of the global state inference model, which can extract essential features from such high-dimensional inputs.

**Ablation Study for prior network**   We conduct the ablation study for the prior network $p_\phi$ by removing the KL constraint in Equation 9 and prior network $p_\phi$ in training process, which is named `GlobeDiff w/o p`. We conduct experiments on the various maps. The experimental results in the Figure 9 indicate that the performance of GlobeDiff can be effectively enhanced by introducing the prior network.

**Ablation Study for Hyper-parameters**   To study the robustness of GlobeDiff across different hyper-parameters, we conduct the following ablation studies. Specifically, we change the diffusion step $K$ from 1 to 8. The experimental results in the left part of Figure 10 show that the state inference is more accurate with the longer denoising steps. In addition, we conduct ablation studies for the model parameters of the U-Net with various residual blocks. The experimental results in the right right of Figure 10 show that the model's capacity has a relatively minor impact on the algorithm's performance. We only need a small model to achieve accurate global state inference.

## 6   CONCLUSION

In this paper, we study the partial observability problem in multi-agent systems. We first propose a generative model-based global state inference framework under two scenarios. Then, we propose the Global State Diffusion Algorithm (GlobeDiff), which formulates the state inference process as a multi-modal diffusion process. The theoretical analysis shows that the estimation error of GlobeDiff can be bounded. Extensive experiments demonstrate that GlobeDiff can not only accurately infer the global state, significantly enhance the algorithm's performance, and be easily integrated with current MARL algorithms. In the future, we will apply our algorithm to real-world tasks to address the challenges of partial observability in real environments.

## REPRODUCIBILITY STATEMENT

We have provided the source code in the supplementary materials, which will be made public after the paper is accepted. We have provided theoretical analysis in the Appendix B. We have also provided implementation details in the Appendix D.

## ACKNOWLEDGMENTS

This work was supported in part by the National Natural Science Foundation of China under Grant 62192751, in part by the 111 International Collaboration Program of China under Grant B25027, in part by the in part by the InnoHK Initiative, The Government of HKSAR, and in part by the Laboratory for AI-Powered Financial Technologies. This work was also supported by the National Key R&D Program of China (No.2022ZD0116405).

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

# A ALGORITHM

---
**Algorithm 1** Global State Diffusion Process

---
1: Initialize the parameters of MARL algorithm and the diffusion model
2: Add offline data to online buffer $\mathcal{D}$
3: **for** each episode **do**
4:    **for** $t \leftarrow 1$ **to** $T$ **do**
5:       Obtain local observation $\boldsymbol{o}_t$
6:       Construct auxiliary state $x_t$
7:       Calculate the encoded latent variables $z$ based on $p_\phi(z \mid x_t)$
8:       Infer the global state $s_t^0$ based on the Eq. 11
9:       Each agent makes decision based on the inferred state $s_t^0$
10:      Send $\boldsymbol{a}_t$ to environment and receive $\boldsymbol{o}_{t+1}, s_{t+1}, r_t$
11:      Store transitions in replay buffer $\mathcal{D}$
12:    **end for**
13:    Update MARL algorithm
14:    **if** Update Global State Diffusion Model **then**
15:       Update global state diffusion model based on the Eq. 9
16:    **end if**
17: **end for**
18: **end**

---

## B Theoretical Analysis

### B.1 Error Bound Analysis for Single Samples with Latent Variable

**Theorem.** *1 (Single-Sample Expectation Error Bound with Latent Variable) Assume the trained model satisfies the following two assumptions. (1) Diffusion noise prediction MSE: $\mathbb{E}_{s^k,x,z,k}[\|\epsilon_\theta(s^k, x, z, k) - \epsilon\|^2] \leq \delta^2$, (2) Prior alignment: $D_{KL}(p_\phi(z \mid x)\|p(z \mid x)) \leq \varepsilon_{KL}$. Then, for any generated sample $\hat{s} \sim p_{\theta,\phi}(s \mid x) = \int p_\theta(s \mid x, z)p_\phi(z \mid x)dz$ and true sample $s \sim p(s \mid x)$, the expected squared error is bounded by:*

$$\mathbb{E}\left[\|\hat{s} - s\|^2\right] \leq 2W_2^2(p_{\theta,\phi}(s \mid x), p(s \mid x)) + 4Var(s \mid x), \tag{14}$$

*where $W_2$ is the 2-Wasserstein distance between $p_{\theta,\phi}(s \mid x)$ and $p(s \mid x)$, $Var(s \mid x) = \mathbb{E}_{p(s|x)}\left[\|s - \mu_{s|x}\|^2\right]$ is the conditional variance and $\mu_{s|x} = \mathbb{E}_{p(s|x)}[s]$ is the conditional mean.*

*Proof.* **Step 1: Error Decomposition via Variance-Bias Tradeoff**
Let $\mu_{s|x} = \mathbb{E}_{p(s|x)}[s]$. For any $\hat{s}$ and $s$, we expand:

$$\|\hat{s} - s\|^2 = \|(\hat{s} - \mu_{s|x}) - (s - \mu_{s|x})\|^2. \tag{15}$$

By the triangle inequality and Young's inequality:

$$\|\hat{s} - s\|^2 \leq 2\|\hat{s} - \mu_{s|x}\|^2 + 2\|s - \mu_{s|x}\|^2. \tag{16}$$

Taking expectation:

$$\mathbb{E}\left[\|\hat{s} - s\|^2\right] \leq 2\mathbb{E}\left[\|\hat{s} - \mu_{s|x}\|^2\right] + 2\mathbb{E}\left[\|s - \mu_{s|x}\|^2\right]. \tag{17}$$

**Step 2: Bounding the First Term via Wasserstein Distance**
The first term represents the deviation of the generated sample from the true conditional mean. By properties of the Wasserstein distance:

$$\mathbb{E}_{p_{\theta,\phi}(s|x)}\left[\|\hat{s} - \mu_{s|x}\|^2\right] \leq W_2^2(p_{\theta,\phi}(s \mid x), \delta_{\mu_{s|x}}) \leq W_2^2(p_{\theta,\phi}(s \mid x), p(s \mid x)) + W_2^2(p(s \mid x), \delta_{\mu_{s|x}}), \tag{18}$$

where $\delta_{\mu_{s|x}}$ is the Dirac delta at $\mu_{s|x}$. The second term equals $Var(s \mid x)$, giving:

$$\mathbb{E}\left[\|\hat{s} - \mu_{s|x}\|^2\right] \leq W_2^2(p_{\theta,\phi}(s \mid x), p(s \mid x)) + Var(s \mid x). \tag{19}$$

**Step 3: Bounding the Second Term**
The second term is exactly the conditional variance:

$$\mathbb{E}\left[\|s - \mu_{s|x}\|^2\right] = Var(s \mid x). \tag{20}$$

**Step 4: Final Synthesis**
Combining all results:

$$\mathbb{E}\left[\|\hat{s} - s\|^2\right] \leq 2\left(W_2^2(p_{\theta,\phi}(s \mid x), p(s \mid x)) + Var(s \mid x)\right) + 2Var(s \mid x) \tag{21}$$

$$= 2W_2^2(p_{\theta,\phi}(s \mid x), p(s \mid x)) + 4Var(s \mid x). \tag{22}$$

which completes the proof. □

### B.2 Connecting Training Loss to Wasserstein Bound with Latent Variable

**Lemma 1.** *Let the following hold. (1) Noise prediction MSE: $\mathbb{E}_{s^k,x,z,k}[\|\epsilon_\theta(s^k, x, z, k) - \epsilon\|^2] \leq \delta^2$, (2) KL divergence: $D_{KL}(p_\phi(z \mid x)\|p(z \mid x)) \leq \varepsilon_{KL}$. Then the Wasserstein-2 distance between $p_{\theta,\phi}(s \mid x)$ and $p(s \mid x)$ is bounded by:*

$$W_2^2(p_{\theta,\phi}(s \mid x), p(s \mid x)) \leq C_1 K\delta^2 + C_2\varepsilon_{KL}, \tag{23}$$

*where $C_1 = \max_k \left(\frac{1-\alpha^k}{\sqrt{\alpha^k(1-\bar{\alpha}^k)}}\right)^2 \prod_{i=k+1}^K (\alpha^i)^{-1}$, $C_2$ is a constant depending on the latent space dimension and geometry, and $K$ is the total number of diffusion steps.*

*Proof.* **Step 1: Single-Step Error Propagation for Diffusion**

The reverse process update at step $k$ conditioned on $z$ is:

$$s^{k-1} = \frac{1}{\sqrt{\alpha^k}} s^k - \frac{1-\alpha^k}{\sqrt{\alpha^k(1-\bar{\alpha}^k)}} \epsilon_\theta(s^k, x, z, k) + \sqrt{\beta^k}\epsilon. \tag{24}$$

The deviation caused by noise prediction error $\Delta\epsilon_k = \epsilon_\theta - \epsilon$ satisfies:

$$\Delta s^{k-1} = \frac{1-\alpha^k}{\sqrt{\alpha^k(1-\bar{\alpha}^k)}} \Delta\epsilon_k + \frac{1}{\sqrt{\alpha^k}} \Delta s^k. \tag{25}$$

**Step 2: Error Accumulation Over $K$ Steps**

Unrolling the error through all $K$ steps:

$$\Delta s^0 = \sum_{k=1}^{K} \left( \prod_{i=k+1}^{K} \frac{1}{\sqrt{\alpha^i}} \right) \frac{1-\alpha^k}{\sqrt{\alpha^k(1-\bar{\alpha}^k)}} \Delta\epsilon_k. \tag{26}$$

Taking the expectation of the squared norm:

$$\mathbb{E}[\|\Delta s^0\|^2] = \mathbb{E}\left[ \left\| \sum_{k=1}^{K} A_k \Delta\epsilon_k \right\|^2 \right], \tag{27}$$

where $A_k = \left( \prod_{i=k+1}^{K} \frac{1}{\sqrt{\alpha^i}} \right) \frac{1-\alpha^k}{\sqrt{\alpha^k(1-\bar{\alpha}^k)}}$.

Expanding the square:

$$\mathbb{E}[\|\Delta s^0\|^2] = \sum_{k=1}^{K} \|A_k\|^2 \mathbb{E}[\|\Delta\epsilon_k\|^2] + 2 \sum_{1 \leq k < l \leq K} \mathbb{E}[\langle A_k \Delta\epsilon_k, A_l \Delta\epsilon_l \rangle]. \tag{28}$$

Assuming the noise prediction errors at different steps are uncorrelated, the cross terms vanish:

$$\mathbb{E}[\|\Delta s^0\|^2] = \sum_{k=1}^{K} \|A_k\|^2 \mathbb{E}[\|\Delta\epsilon_k\|^2] \leq \delta^2 \sum_{k=1}^{K} \|A_k\|^2. \tag{29}$$

Now, we need to bound $\sum_{k=1}^{K} \|A_k\|^2$. Let:

$$\|A_k\|^2 = \left( \prod_{i=k+1}^{K} \frac{1}{\alpha^i} \right) \left( \frac{1-\alpha^k}{\sqrt{\alpha^k(1-\bar{\alpha}^k)}} \right)^2. \tag{30}$$

Let $C_1 = \max_k \left( \prod_{i=k+1}^{K} \frac{1}{\alpha^i} \right) \left( \frac{1-\alpha^k}{\sqrt{\alpha^k(1-\bar{\alpha}^k)}} \right)^2$. Then:

$$\sum_{k=1}^{K} \|A_k\|^2 \leq C_1 K, \tag{31}$$

Thus:

$$\mathbb{E}[\|\Delta s^0\|^2] \leq C_1 K \delta^2. \tag{32}$$

**Step 3: Incorporating KL Divergence Error**

The KL divergence bound $\varepsilon_{\text{KL}}$ ensures that the prior $p_\phi(z \mid x)$ is close to the true posterior $p(z \mid x)$. By the data processing inequality for Wasserstein distance:

$$W_2^2\left( \int p_\theta(s \mid x, z) p_\phi(z \mid x) dz, \int p(s \mid x, z) p(z \mid x) dz \right) \leq C_2 W_2^2(p_\phi(z \mid x), p(z \mid x)), \quad (33)$$

where $C_2$ depends on the Lipschitz constant of the mapping $z \mapsto p(s \mid x, z)$. The constant $C_2$ depends on the latent space dimension for the following reasons. (1) Dimension Scaling of Wasserstein Distance: For distributions in $\mathbb{R}^{d_z}$, the Wasserstein distance typically scales with $\sqrt{d_z}$ due to concentration of measure phenomena. This is known as the "curse of dimensionality" in optimal transport. (2) Talagrand's Inequality: If $p(z \mid x)$ is log-concave (e.g., Gaussian), then Talagrand's inequality gives:

$$W_2^2(p_\phi(z \mid x), p(z \mid x)) \leq 2C_{\text{TI}} D_{\text{KL}}(p_\phi(z \mid x) \| p(z \mid x)), \tag{34}$$

where $C_{\text{TI}}$ is the Poincaré constant of $p(z \mid x)$. For isotropic Gaussians in $\mathbb{R}^{d_z}$, this constant scales as $O(d_z)$. (3) Lipschitz Constant: The mapping $z \mapsto p(s \mid x, z)$ typically has a Lipschitz constant that grows with the dimension $d_z$ due to the increased complexity of the conditional distribution. Thus, we can write:

$$W_2^2(p_\phi(z \mid x), p(z \mid x)) \leq C_2' d_z \varepsilon_{\text{KL}}, \tag{35}$$

where $C_2'$ is a dimension-independent constant.

**Step 4: Combined Bound**
Combining both error sources using the triangle inequality for Wasserstein distance:

$$W_2^2(p_{\theta,\phi}(s \mid x), p(s \mid x)) \leq 2W_2^2(p_{\theta,\phi}(s \mid x), p(s \mid x, z)p(z \mid x)) + 2W_2^2(p(s \mid x, z)p(z \mid x), p(s \mid x)) \tag{36}$$

$$\leq 2C_1 K \delta^2 + 2C_2' d_z \varepsilon_{\text{KL}}. \tag{37}$$

Absorbing constants into $C_1$ and $C_2'$. Denote $C_2'$ as $C_2$, and noting that $C_2'$ depends on $d_z$, we obtain the final bound:

$$W_2^2(p_{\theta,\phi}(s \mid x), p(s \mid x)) \leq C_1 K \delta^2 + C_2 \varepsilon_{\text{KL}}. \tag{38}$$

$\square$

### B.3 ONE-TO-MANY MAPPING CASE WITH LATENT VARIABLE

**Theorem.** *2 (Multi-Modal Error Bound with Latent Variable) Under the following conditions: (1) The true conditional distribution $p(s \mid x) = \sum_{i=1}^N w_i \mathcal{N}(s; \mu_i(x), \Sigma_i(x))$ has $N$ modes with minimum inter-mode distance $D = \min_{i \neq j} \|\mu_i(x) - \mu_j(x)\| \geq 2\sqrt{d}$. (2) Mode separation condition: $D > 4\sqrt{C_1 K \delta^2 + C_2 \varepsilon_{KL} + \max_i Tr(\Sigma_i(x))}$ (3) The model satisfies $\mathbb{E}[\|\epsilon_\theta - \epsilon\|^2] \leq \delta^2$ and $D_{KL}(p_\phi(z \mid x) \| p(z \mid x)) \leq \varepsilon_{KL}$. Then, for any generated sample $\hat{s} \sim p_{\theta,\phi}(s \mid x)$, there exists a mode $\mu_j(x)$ such that:*

$$\mathbb{E}\left[\|\hat{s} - \mu_j(x)\|^2\right] \leq C_1 K \delta^2 + C_2 \varepsilon_{KL} + 2 \max_i Tr(\Sigma_i(x)) + \mathcal{O}\left(e^{-D^2/(8\sigma_{max}^2)}\right), \tag{39}$$

*where $\sigma_{max}^2 = \max_i Tr(\Sigma_i(x))$, and $C_1, C_2$ are constants depending on the diffusion scheduler and latent space geometry.*

*Proof.* **Step 1: Voronoi Partitioning and Projection Operator**
The state space $\mathbb{R}^d$ is partitioned into $N$ Voronoi regions $\{V_i\}_{i=1}^N$ centered at the mode centers $\{\mu_i(x)\}$. Define the projection operator:

$$\phi(s) = \sum_{i=1}^N \mu_i(x) \cdot \mathbf{1}_{\{s \in V_i\}}, \tag{40}$$

which maps any point $s$ to the center of its containing Voronoi region.

**Step 2: Conditional Distribution Definitions**
For each Voronoi region $V_i$, define the conditional distributions:

- True state conditional distribution: $p_i(s) = p(s \mid x, s \in V_i)$

- Generated state conditional distribution: $q_i(\hat{s}) = p_{\theta,\phi}(\hat{s} \mid x, \hat{s} \in V_i)$

Under the mode separation condition, the true state conditional distribution $p_i(s)$ approximates unimodel Gaussian distribution $\mathcal{N}(\mu_i(x), \Sigma_i(x))$ with exponentially small error:

$$W_2^2(p_i(s), \mathcal{N}(\mu_i(x), \Sigma_i(x))) \leq \mathcal{O}\left(e^{-D^2/(8\sigma_{\max}^2)}\right). \tag{41}$$

**Step 3: Per-Region Projection Error Bound**
For each Voronoi region $V_i$, consider the conditional expectation of the projection error:

$$\mathbb{E}[\|\hat{s} - \phi(\hat{s})\|^2 \mid \hat{s} \in V_i] = \mathbb{E}[\|\hat{s} - \mu_i(x)\|^2 \mid \hat{s} \in V_i]. \tag{42}$$

Using the triangle inequality for Wasserstein distance:

$$\mathbb{E}[\|\hat{s} - \mu_i(x)\|^2 \mid \hat{s} \in V_i] = W_2^2(q_i, \delta_{\mu_i(x)}) \tag{43}$$

$$\leq \left(W_2(q_i, p_i) + W_2(p_i, \delta_{\mu_i(x)})\right)^2 \tag{44}$$

$$\leq 2W_2^2(q_i, p_i) + 2W_2^2(p_i, \delta_{\mu_i(x)}), \tag{45}$$

where $\delta_{\mu_i(x)}$ is the Dirac delta distribution at $\mu_i(x)$. For the first term, according to Llama1, we have:

$$W_2^2(q_i, p_i) \leq \frac{W_2^2(p_{\theta,\phi}(s \mid x), p(s \mid x))}{P(\hat{s} \in V_i)} + \mathcal{O}\left(e^{-D^2/(8\sigma_{\max}^2)}\right) \leq \frac{C_1 K \delta^2 + C_2 \varepsilon_{\mathrm{KL}}}{P(\hat{s} \in V_i)} + \mathcal{O}\left(e^{-D^2/(8\sigma_{\max}^2)}\right). \tag{46}$$

And for the second term, we have:

$$W_2^2(p_i, \delta_{\mu_i(x)}) = \mathbb{E}_{p_i}[\|s - \mu_i(x)\|^2] \leq \mathrm{Tr}(\Sigma_i(x)) + \mathcal{O}\left(e^{-D^2/(8\sigma_{\max}^2)}\right) \leq \sigma_{\max}^2 + \mathcal{O}\left(e^{-D^2/(8\sigma_{\max}^2)}\right). \tag{47}$$

**Step 5: Aggregation Over All Regions**
Compute the global expectation:

$$\mathbb{E}[\|\hat{s} - \phi(\hat{s})\|^2] = \sum_{i=1}^{N} P(\hat{s} \in V_i) \cdot \mathbb{E}[\|\hat{s} - \phi(\hat{s})\|^2 \mid \hat{s} \in V_i] \tag{48}$$

$$\leq \sum_{i=1}^{N} P(\hat{s} \in V_i) \left[2\left(\frac{C_1 K \delta^2 + C_2 \varepsilon_{\mathrm{KL}}}{P(\hat{s} \in V_i)} + \mathcal{O}\left(e^{-D^2/(8\sigma_{\max}^2)}\right)\right) + 2\left(\sigma_{\max}^2 + \mathcal{O}\left(e^{-D^2/(8\sigma_{\max}^2)}\right)\right)\right] \tag{49}$$

$$= \sum_{i=1}^{N} \left[2(C_1 K \delta^2 + C_2 \varepsilon_{\mathrm{KL}}) + 2P(\hat{s} \in V_i)\sigma_{\max}^2 + \mathcal{O}\left(e^{-D^2/(8\sigma_{\max}^2)}\right)\right] \tag{50}$$

$$= 2N(C_1 K \delta^2 + C_2 \varepsilon_{\mathrm{KL}}) + 2\max_i \mathrm{Tr}(\Sigma_i(x)) + \mathcal{O}\left(e^{-D^2/(8\sigma_{\max}^2)}\right). \tag{51}$$

Redefining constants $C_1' = 2NC_1$ and $C_2' = 2NC_2$, and absorbing factors:

$$\mathbb{E}[\|\hat{s} - \phi(\hat{s})\|^2] \leq C_1 K \delta^2 + C_2 \varepsilon_{\mathrm{KL}} + 2\max_i \mathrm{Tr}(\Sigma_i(x)) + \mathcal{O}\left(e^{-D^2/(8\sigma_{\max}^2)}\right). \tag{52}$$

When $D \gg \sigma_{\max}$, the exponential term becomes negligible.

$\square$

## C  ADDITIONAL EXPERIMENTS

### C.1  QUANTIFYING THE DIVERSITY

We conducted additional experiments to illustrate the semantic representation learned by $z$. Specifically, in the SMAC-v2 Zerg 5v5 environment, we collected observational data across all timesteps and identified both the true global states and the global states generated by GlobeDiff. These state samples were then projected into a two-dimensional plane using t-SNE for visualization, and a 3D density surface was estimated using Kernel Density Estimation (KDE). The experimental results in Figure 11 show that the resulting 3D plots illustrate the normalized probability density ($z$-axis) over the 2D t-SNE embedding ($x$- and $y$-axis).

In the leftmost subplot, we observe that the same observational input often corresponds to multiple real global state clusters at distinct locations in the 2D plane, forming several prominent Gaussian modes. The latent variable $z$ can thus be interpreted as a positional encoding for the coordinates of these Gaussian modes in the embedded space. The three subplots on the right display the global states generated by three selected agents, which faithfully reconstruct the multi-modal characteristics of the true state distribution. Notably, the location, shape, and amplitude of the reconstructed Gaussian modes closely resemble those of the real state distribution, empirically demonstrating $z$'s role in capturing semantic variations.

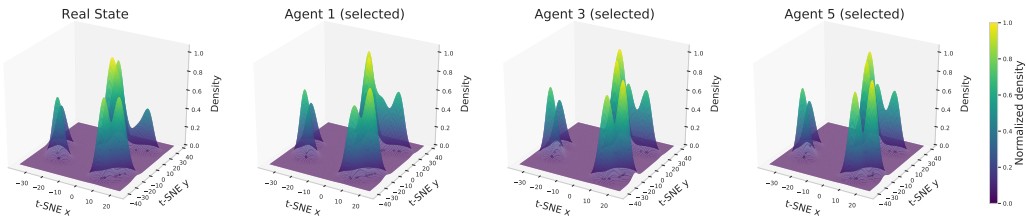

Figure 11: Experiments of the qualitative analysis.

### C.2  EXPERIMENTS UNDER SAME PARAMETER COUNT

We conducted a new experiment where we increased the parameter count of the vanilla MAPPO networks to match the total number of parameters in GlobeDiff (which includes both the diffusion model and the RL policy). We calculated the total parameters in GlobeDiff to be approximately 12–14M. Accordingly, we expanded both the MAPPO actor and critic networks to four-layer MLPs with 2048 hidden units per layer, creating a "Vanilla MAPPO (Large)" baseline with a comparable parameter budget (approx. 13.5–14M parameters).

The experimental results in Table 1 show that despite this significant increase in capacity, the performance gain of Vanilla MAPPO (Large) remains limited. These results suggest that merely increasing the capacity of a standard recurrent policy is insufficient for it to implicitly learn the complex task of multi-modal global state reconstruction, highlighting the necessity of an explicit generative module like GlobeDiff.

|              | Vanilla MAPPO | Vanilla MAPPO (Large) | GlobeDiff       |
| ------------ | ------------- | --------------------- | --------------- |
| zerg 5v5     | 0.22±0.01     | 0.23±0.00             | 0.33±0.02       |
| protoss 5v5  | 0.21±0.02     | 0.24±0.01             | 0.38±0.01       |
| terran 5v5   | 0.16±0.01     | 0.17±0.00             | 0.24±0.01       |
| zerg 10v10   | 0.13±0.01     | 0.15±0.01             | 0.25±0.01       |
| zerg 10v11   | 0.06±0.01     | 0.07±0.01             | 0.12±0.01       |
| terran 10v11 | 0.02±0.00     | 0.03±0.00             | 0.07±0.01       |
| MMM2         | 0.27±0.08     | 0.01±0.01             | 0.49±0.11       |
| 3s5z vs 3s6z | 0.20±0.01     | 0.01±0.01             | 0.28±0.04       |
| 6h vs 8z     | 0.12±0.01     | 0.01±0.00             | 0.47±0.04       |

Table 1: Comparison results between Vanilla MAPPO, Vanilla MAPPO (Large), and GlobeDiff

## D  IMPLEMENTATION DETAILS

In practical implementation, to mitigate the data bias in the offline dataset and ensure the effective state inference, we update the GlobeDiff during online learning with an update interval of 50 episodes. Under the CTDE framework, we select the historical observation length $m = 3$. Please refer to Table 2, Tables 3 and 4 for the detailed hyper-parameters.

The architecture of the Diffusion model consists of a U-Net structure with two repeated residual blocks, as shown in Figure 12. Each block consisted of two temporal convolutions, followed by group norm (Wu & He, 2018), and a final Mish nonlinearity (Misra, 2019). The U-Maze dim refers to the multiplicative factor that reduces the output dimension relative to the input dimension during the down-sampling process.

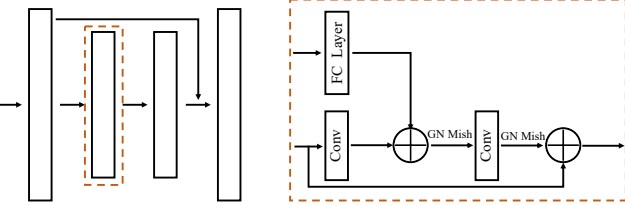

Figure 12: U-Net architecture in Diffusion model.

Table 2: Hyper-parameters for MAPPO.

| Hyper-parameters | Value | Hyper-parameters | Value | Hyper-parameters | Value |
|---|---|---|---|---|---|
| Critic Learning Rate | 5e-4 | Actor Learning Rate | 5e-4 | Use GAE | True |
| Gain | 0.01 | Optim | 1e-5 | Batch Size | 64 |
| Training Threads | 4 | Num Mini-Batch | 1 | Rollout Threads | 8 |
| Entropy Coef | 0.01 | Max Grad Norm | 10 | Episode Length | 400 |
| Optimizer | Adam | Hidden Layer Dim | 64 | GAE $\lambda$ | 0.95 |
| Activation Function | Relu | PPO Epoch | 15 | $\gamma$ | 0.99 |

Table 3: Hyper-parameters for prior network $p_\phi$ and posterior network $q_\psi$.

| Prior Network $p_\phi$ | Value | Posterior Network $q_\psi$ | Value |
|---|---|---|---|
| $z$ dim | 16 | $z$ dim | 16 |
| Hidden Layer Dim | 1024 | Hidden Layer Dim | 1024 |
| Hidden Layer Num | 3 | Hidden Layer Num | 3 |
| Activation Function | Relu | Activation Function | Relu |
| Learning Rate | 2e-4 | Learning Rate | 2e-4 |
| Weight Decay | 1e-4 | Weight Decay | 1e-4 |
| Batch Size | 32 | Batch Size | 32 |

Table 4: Hyper-parameters for Generative methods.

| GlobeDiff | Value | VAE | Value | MLP | Value |
|---|---|---|---|---|---|
| Down-sampling Factor | 8 | Latent Dim | 256 | Hidden Layer Dim | 1024 |
| Diffusion Steps | 5 | Hidden Layer Dim | 1024 | Hidden Layer Num | 4 |
| Residual Blocks | 2 | Activation Function | Relu | Activation Function | Relu |
| Learning Rate | 2e-4 | Learning Rate | 1e-4 | Learning Rate | 3e-4 |
| Batch Size | 32 | Batch Size | 32 | Batch Size | 32 |

