# OpenReview forum: "GlobeDiff: State Diffusion Process for Partial Observability in Multi-Agent System"
_ICLR.cc/2026/Conference — ICLR 2026 Poster_

### Official Review · Reviewer_TZGo · 2025-10-23

**Soundness:** 2
**Presentation:** 2
**Contribution:** 2
**Rating:** 4
**Confidence:** 4

**Summary:**

The paper introduces the idea of using diffusion models to infer full state observations from local (single agent) observations in general partially observable multi-agent reinforcement learning problems. They provide some analysis of how to cast the learning problem in terms of the generative models, they derive theoretically some error bounds in the expected prediction errors between the generative model and the true fully observable state, and include a wide range of empirical results demonstrating how incorporating a diffusion model into a baseline MARL algorithm helps improve significantly the average returns obtained by the agents.

**Strengths:**

- The problem is highly relevant. Partial observability in dec-POMDPs is a prevalent bottleneck in terms of general MARL algorithms succeeding at producing optimal distributed policies in complex MARL environments.
- The solution the authors propose is reasonable and very straightforward: use powerful generative models to aid agents form posterior densities over true fully observable states based on local observations.
- The empirical results seem thorough, and the authors compare against a reasonable amount of baselines.

**Weaknesses:**

- Using diffusion models to approximate posterior distributions of fully observable states in Dec-POMDPs is not really a novel idea. See [1]; authors even test MAPPO implementations with diffusion generative models in the SMAC benchmarks.
- Theorem 1 does not really add any insight or seems to be particularly useful; it states that the expected approximation error between true samples and generated ones is bounded by a linear combination of some distance between distributions and the variance of the generative model (this is not a very insightful result). The proof also follows almost immediately from the triangle inequality.
- I am slightly concerned of the validity of the baseline comparisons. I am not convinced it is fair to compare GlobeDiff+MAPPO against a vanilla MAPPO with the same number of trainable parameters. See below for a question on this.
- Some sentences in the introduction are not very clear. First, I would appreciate some nuance in the assumption that a MARL problem is necessarily partially observable (this is not the case in general, there are game theoretic problems that are fully observable like eg bimatrix games...). Also, the following sentence (lines 49-51) is not clear: "Unfortunately, the focus on interpreting past experiences without significantly enhancing the agent's ability to leverage global information". Isn't interpreting past experiences a way of inferring global information?
- (*Minor but I had to raise it*) In the paper title, as well as in the first line of the introduction, 'multi-agent system' should be plural, 'multi-agent systems', and the first line should be 'Multi-agent systems have...'.


[1] Wang, Tonghan, et al. "On Diffusion Models for Multi-Agent Partial Observability: Shared Attractors, Error Bounds, and Composite Flow." Proceedings of the 24th International Conference on Autonomous Agents and Multiagent Systems. 2025.

**Questions:**

1. Can the authors reference [1] above and frame their work in this context? I believe the author's experimental results are more thorough than [1], but I believe it affects the claims that diffusion models have not been used in Dec-POMDPs.
2. Is theorem 1 a contribution of equal weight as Theorem 2? I am not sure it's a relevant result, I believe more space dedicated to Theorem 2 would probably benefit the paper.
3. In eq (4) authors derive the ELBO of a generative model, which is a well known expression and a common approach to training variational approximations to posterior densities in general Bayesian generative problems. A reference to this and some nuance would make the paper clearer.
4. I have a fundamental question about the baselines used to compare. In my view, GlobeDiff+MAPPO is in some way adding a large amount of model complexity to vanilla MAPPO. While there is no good way of measuring this and ensuring it's constant among all baselines, I don't believe it is fair to compare baselines against each-other without making any reference to the size of the policy/value networks. Were the architectures of the policies kept constant across all baselines? I would then argue that it is trivial that GlobeDiff+MAPPO provides a performance increase, since at the very least this is adding model complexity to the representation the agents learn. What I would be interested in knowing is, if for example, one can train GlobeDiff+MAPPO where the number of parameters in the networks (of the diffusion model + the RL policy) is **the same** as the number of parameters in a vanilla MAPPO policy. If even under comparable model complexity the authors observe such improvements, I will be more convinced that the approach adds any value to the MARL field. If under a similar (total) number of parameters the performance is similar, I would argue perhaps deep RL recurrent policies with enough complexity generate some form of internal representation of the global state reconstruction based on agent observations (which is, in fact, the implicit argument behind using recurrent policies in Dec-POMDPs).

---

> ### Author Response · Authors · 2025-11-21
> **Response to Reviewer TZGo (I)**
>
> Dear Reviewer,
>
> Thank you for your valuable and constructive comments. We have performed additional experiments and analysis to address your concerns, and we believe the following responses and the corresponding revisions have significantly strengthened our manuscript.
>
> **W1 and Q1: Related work of using diffusion models to approximate posterior distributions of fully observable states in Dec-POMDPs. Can the authors reference [1] and frame their work in this context?**
>
> **A for W1 and Q1:**
> Thank you for bringing the relevant work by [1] to our attention. Our work was developed concurrently, and we have now added a citation and discussion acknowledging their contribution in exploring diffusion models for multi-agent partial observability.
>
> While both works employ diffusion models for state reconstruction, our approach diverges fundamentally in its problem formulation and technical innovation:
>
> - Methodological Formulation: The work in [1] focuses on approximating belief distributions through shared attractors, leveraging diffusion as a belief updater within existing policy architectures. In contrast, GlobeDiff introduces a conditional, multi-modal diffusion process that explicitly addresses the "one-to-many" ambiguity in state estimation, where a single local observation can correspond to multiple plausible global states. By modeling the conditional distribution
> $p(s|x)$ with a latent variable $z$, our method is designed to capture complex, multi-modal global state distributions—a key challenge not directly addressed by their composite flow framework.
> - Empirical Evaluation: We extend our evaluation beyond SMAC-v1 to the more challenging SMAC-v2 benchmarks, where partial observability is exacerbated by randomized initial conditions and heterogeneous agent types. Our experiments demonstrate strong performance, validating GlobeDiff’s effectiveness and adaptability in these complex settings.
>
> We have updated the Related Work section to include [1] and have refined our novelty statement to clarify that while diffusion models have been explored for Dec-POMDPs, our formulation of state inference as a conditional, multi-modal generative process represents a distinct and complementary advancement for state reconstruction under partial observability.
>
> **W2 and Q2: What's insights of Theorem 1? Is theorem 1 a contribution of equal weight as Theorem 2?**
>
> **A for W2 and Q2:**
> Thank you for this question, which allows us to clarify the distinct yet complementary roles of our two main theoretical results. Theorem 1 and Theorem 2 are designed to work in tandem.
>
> - Theorem 1 establishes a foundational, general-purpose error bound that holds under broad conditions, without requiring any assumptions about mode separation or data modality. It proves that the reconstruction error is controlled by the distributional distance achieved during training, specifically the Wasserstein distance and the conditional variance of the true state distribution.
>
> - Theorem 2 builds upon this foundation to specifically address multi-modal distributions. By introducing a mode separation condition, it derives a much tighter, mode-level guarantee on the distance from a generated sample to its nearest true mode center. When modes are well-separated, each generated sample can be confidently assigned to a single true mode, leading to a tight per-mode bound with an exponentially small misassignment term.
>
> In essence, Theorem 1 offers a universal error bound, while Theorem 2 provides a stronger, more specialized guarantee for the multi-modal settings our method is designed for. We have revised the paper to better articulate this symbiotic relationship and ensure both theorems are presented with appropriate context and weight.

---

> ### Author Response · Authors · 2025-11-21
> **Response to Reviewer TZGo (II)**
>
> **W3 and Q3: Were the architectures of the policies kept constant across all baselines? Additional experiments of training GlobeDiff+MAPPO where the number of parameters in the networks (of the diffusion model + the RL policy) is the same as the number of parameters in a vanilla MAPPO policy.**
>
> **A for W3:**
> We confirm that the RL policy architectures were kept consistent across all algorithms; that is, all agents were configured with identical parameters and network sizes for the underlying MAPPO policy.
>
> Furthermore, as suggested, we conducted a new experiment where we increased the parameter count of the vanilla MAPPO networks to match the total number of parameters in GlobeDiff (which includes both the diffusion model and the RL policy). We calculated the total parameters in GlobeDiff to be approximately 12–14M. Accordingly, we expanded both the MAPPO actor and critic networks to four-layer MLPs with 2048 hidden units per layer, creating a "Vanilla MAPPO (Large)" baseline with a comparable parameter budget (approx. 13.5–14M parameters).
>
> The experimental results in Table 1 show that despite this significant increase in capacity, the performance gain of Vanilla MAPPO (Large) remains limited. These results suggest that merely increasing the capacity of a standard recurrent policy is insufficient for it to implicitly learn the complex task of multi-modal global state reconstruction, highlighting the necessity of an explicit generative module like GlobeDiff.
>
> We have added these new experimental results to Appendix C in the revised manuscript.
>
> | | Vanilla MAPPO | Vanilla MAPPO (Large) | GlobeDiff |
> |:-:|:-:|:-:|:-:|
> |zerg 5v5|0.22$\pm$0.01 |0.23$\pm$0.00|0.33$\pm$0.02
> |protoss 5v5|0.21$\pm$0.02 |0.24$\pm$0.01|0.38$\pm$0.01
> |terran 5v5| 0.16$\pm$0.01 |0.17$\pm$0.00|0.24$\pm$0.01
> |zerg 10v10| 0.13$\pm$0.01|0.15$\pm$0.01|0.25$\pm$0.01
> |zerg 10v11| 0.06$\pm$0.01|0.07$\pm$0.01|0.12$\pm$0.01
> |terran 10v11|  0.02$\pm$0.00|0.03$\pm$0.00|0.07$\pm$0.01
> |MMM2| 0.27$\pm$0.08|0.01$\pm$0.01|0.49$\pm$0.11
> |3s5z vs 3s6z| 0.20$\pm$0.01|0.01$\pm$0.01|0.28$\pm$0.04
> |6h vs 8z| 0.12$\pm$0.01|0.01$\pm$0.00|0.47$\pm$0.04
>
> Table 1. Comparison results between Vanilla MAPPO, Vanilla MAPPO (Large), and GlobeDiff.
>
> **W4: Some sentences in the introduction are not very clear.**
>
> **A for W4:**
> Thank you for this feedback. We have substantially rewritten the introduction to improve clarity, flow, and impact, ensuring our core contributions are articulated more forcefully.
>
> **W5: Description of multi-agent systems.**
>
> **A for W5:**
> Thank you for noting this. We have corrected the description in the revised version.
>
>
> **Q3: A reference to the ELBO in eq (4)  and some nuance would make the paper clearer.**
>
> **A for Q3:**
> Thank you for this excellent suggestion. We have now added a reference to the ELBO and clarified its connection to our objective function in Equation (4) in the revised manuscript, which we agree makes the formulation clearer.
>
> Thank you again for your thorough and insightful feedback, which has been instrumental in improving our work. We hope our detailed responses and the substantial revisions to the manuscript have fully addressed your concerns.
>
> **Reference**
>
> [1] Wang, Tonghan, et al. "On Diffusion Models for Multi-Agent Partial Observability: Shared Attractors, Error Bounds, and Composite Flow." Proceedings of the 24th International Conference on Autonomous Agents and Multiagent Systems. 2025.

---

> > ### Comment · Reviewer_TZGo · 2025-11-25
> > **Acknowledgement**
> >
> > Dear Authors
> >
> > Thank you for the thorough rebuttal. I'm relatively satisfied with the updated state of the paper and will increase my score to reflect this.

---

> ### Author Response · Authors · 2025-11-25
> **Thanks for raising the score!**
>
> Dear Reviewer
>
> We would like to thank the reviewer for raising the score! We sincerely appreciate your time, valuable comments, and thoughtful engagement throughout the review process, which helped us significantly improve the paper's strengths. We will incorporate all feedback in the final version. We are grateful for your support and wish you all the best.

---

### Official Review · Reviewer_mKZR · 2025-10-31

**Soundness:** 3
**Presentation:** 2
**Contribution:** 3
**Rating:** 4
**Confidence:** 4

**Summary:**

This paper proposes GlobeDiff, a method that leverages a conditional diffusion model with a latent variable to infer the global state from local observations in partially observable multi-agent systems. The approach is integrated into the CTDE framework and evaluated on modified SMAC benchmarks. The paper includes theoretical error bounds and extensive experimental results showing superior performance over several baselines.

While the core idea of using a generative model for state inference is novel in the MARL/PO context and the results are promising, the paper's presentation significantly undersells its contributions. The writing often obscures the key insights and motivations, making the work appear more incremental than it likely is.

**Strengths:**

Formulating global state inference as a conditional, multi-modal generation problem is a compelling and timely approach. The use of a diffusion model to capture the "one-to-many" mapping inherent in PO is a significant conceptual contribution beyond simple belief estimation.

The method is well-designed. The introduction of the latent variable z to handle multi-modality, coupled with the specific variational training objective (Eq. 10) that aligns the prior and posterior networks, is a non-trivial and thoughtful adaptation of the diffusion model framework for this problem.

The experimental results are comprehensive. GlobeDemonstrates consistent and significant performance gains across a wide range of challenging SMAC (PO) scenarios against a diverse set of baselines (belief-based, communication-based, and other generative models).

Providing theoretical error bounds (Theorems 1 & 2) for both unimodal and multi-modal cases is a notable strength that adds rigor and credibility to the proposed approach.

The ablation studies (prior network, diffusion steps) effectively validate key design choices. The state reconstruction visualization using t-SNE and Voronoi polygons (Fig. 5) is excellent and provides clear, intuitive evidence of the model's capability.

**Weaknesses:**

（1） The introduction successfully outlines the problem but fails to clearly and forcefully state the paper's core intellectual contribution. The description of GlobeDiff is presented as a method, not a solution to a fundamental challenge (the "one-to-many" mapping). The key insight—that generative models can sample from the distribution of plausible global states while discriminative methods collapse to a single mode—is buried in the methodology rather than being the central thesis of the paper.

（2） The method section dives too quickly into mathematical derivations without first providing a high-level, intuitive overview of the framework. The reader is left to reverse-engineer the design philosophy from the equations. A paragraph explaining the roles of z as a "mode selector" and the loss function as a way to bridge the training-inference gap for z is crucial.

（3） The analysis largely stops at reporting that GlobeDiff performs better. It lacks a deep, diagnostic analysis of why and when it excels.

（4） There is no qualitative analysis of specific episodes or timesteps. For example, showing a situation where a local observation is ambiguous, and illustrating the different global states sampled by GlobeDiff versus the incorrect or averaged state estimated by a baseline (like Dynamic Belief) would be immensely powerful.

（5） The paper's central claim is handling multi-modality, but this is not directly tested or contrasted. Quantifying the diversity of generated states (e.g., using metrics like Average Distance to Nearest Neighbor within a batch of samples for the same x) and comparing it to the diversity of the true underlying states would provide strong, direct evidence.

（6） Attributing VAE/MLP's poorer performance merely to "limited representation capacity" is insufficient. A discussion, supported by visualization, showing that the VAE produces blurry "averaged" states would directly support the multi-modal argument.

（7） The computational overhead of running a multi-step denoising process during execution is not discussed. This is a critical consideration for real-time applications.

（8） A discussion of the method's limitations (e.g., performance in environments with extremely high-dimensional state spaces, or when the "one-to-many" mapping is exceptionally complex) would provide a more balanced view and guide future research.

**Questions:**

The latent variable z is central to the method. Can you provide more intuition or analysis of what z learns to represent? Does it correspond to semantically meaningful aspects of the global state?

The communication scenario (Eq. 2) is presented but not analyzed. How critical is the quality and scope of the communicated information to the performance of GlobeDiff? Have you tested scenarios with limited or noisy communication?

Theorem 2 requires a "mode separation condition." How often do you expect this condition to hold in practice, and what is the empirical performance of GlobeDiff when the modes of the global state distribution are less well-separated?

---

> ### Author Response · Authors · 2025-11-21
> **Response to Reviewer mKZR (I)**
>
> Dear Reviewer,
>
> Thank you for your valuable and constructive comments. We have performed additional experiments and analysis to address your concerns, and we believe the following responses have significantly strengthened our manuscript.
>
> **W1: The introduction section should forcefully state the paper's core intellectual contribution.**
>
> **A for W1:**
> Thank you for this insightful comment. As you suggested, we have thoroughly rewritten the introduction section to more clearly and forcefully articulate the paper's core intellectual contribution, particularly emphasizing the "one-to-many" mapping problem and our generative solution.
>
> **W2: The method section should first provides a high-level, intuitive overview of the framework.
> Additional explanation for the roles of z is crucial.**
>
> **A for W2:**
> We appreciate this suggestion. As requested, we have added a high-level, intuitive overview at the beginning of the Methods section in the revised paper. This overview includes a dedicated paragraph explaining the crucial role of the latent variable
> $z$ as a mode selector and how our loss function bridges the training-inference gap for $z$, thereby making the manuscript clearer and more accessible.
>
> **W3: The analysis lacks a deep, diagnostic analysis of why and when GlobeDiff excels.**
>
> **A for W3:**
> Thank you for this important feedback. We have conducted a more in-depth and diagnostic analysis of why and when GlobeDiff excels, covering the following aspects:
>
> - Theoretical Analysis: We provide a theoretical analysis for GlobeDiff by proving a tight upper error bound for the generated states. Specifically, Theorem 1 establishes a general error bound under broad conditions, while Theorem 2 builds upon this by addressing multi-modal distributions. These theorems elucidate GlobeDiff's strong performance: it benefits from both general distributional guarantees and tighter, mode-specific bounds when distinct modes are sufficiently separated, ensuring high-quality generation with minimal bias.
> - Additional Experiments: We conducted additional experiments comparing GlobeDiff with baselines. As detailed in **A for W4 and W6** and **A for W5 and Q1**, these experiments confirm that the true global states reconstructed from partial observations indeed exhibit a multi-modal distribution. GlobeDiff effectively models this multi-modal characteristic, whereas alternative approaches like VAEs and MLPs fail to do so. These visualizations further validate the rationale behind using the latent variable $z$ to capture multi-modality.
>
> We have incorporated the above analysis into the revised version to provide a more comprehensive understanding of GlobeDiff.
>
> **W4 and W6: Qualitative analysis of specific episodes or timesteps.
> Visualization results are necessary to support the multi-modal argument for VAE/MLP.**
>
> **A for W4 and W6:**
> In this work, our baselines fall into two categories: belief-based methods (e.g., LBS, Dynamic Belief, CommFormer) and generative methods (VAE, MLP). Belief-based methods learn a belief model to approximate hidden information, which is then concatenated with raw observations for decision-making. Therefore, directly visualizing the belief information generated by these methods is challenging.
>
> In contrast, generative methods directly infer the global state from raw observations. Consequently, we focused on visualizing the global states inferred by GlobeDiff, VAE, and MLP. The additional results in Figure 5 of the revised paper demonstrate that throughout the training process, the global states predicted by GlobeDiff become increasingly aligned with the true states. However, the global states predicted by VAE and MLP consistently remain significantly different from the true states, often collapsing to a single mode or producing averaged, non-physical states.

---

> ### Author Response · Authors · 2025-11-21
> **Response to Reviewer mKZR (II)**
>
> **W5 and Q1: Quantifying the diversity of generated states and comparing it to the diversity of the true underlying states.
> Can you provide more intuition or analysis of what $z$ learns to represent? Does it correspond to semantically meaningful aspects of the global state?**
>
> **A for W5 and Q1:**
> In the following response, we first provide an intuitive interpretation of $z$, followed by additional experiments quantifying the diversity of true states versus generated states.
>
> - Intuitive interpretation of $z$: We model the task of inferring the global state from local observations as a multi-modal problem, which can be conceptualized as being represented by a Gaussian mixture distribution. The overall state distribution in the full space is expressed as a superposition of multiple unimodal Gaussian distributions, with this superpositional property reflected in each dimension of the state. Accordingly, the latent variable $z$ can be regarded as a positional encoding for each distinct Gaussian mode. During the inference phase of GlobeDiff, a specific latent variable  $z$ is first selected via the network $p_{\phi}(z|x)$, effectively choosing a particular Gaussian mode. The subsequent conditional sampling process of the diffusion model then generates a global state near the center of the selected Gaussian mode.
>
> - Quantifying the diversity: As suggested, we conducted additional experiments to illustrate the semantic representation learned by $z$. Specifically, in the SMAC-v2 Zerg 5v5 environment, we collected observational data across all timesteps and identified both the true global states and the global states generated by GlobeDiff. These state samples were then projected into a two-dimensional plane using t-SNE for visualization, and a 3D density surface was estimated using Kernel Density Estimation (KDE). The experimental results in Figure 11 in Appendix C of the revised paper show that the resulting 3D plots illustrate the normalized probability density ($z$-axis) over the 2D t-SNE embedding ($x$- and $y$-axis).
>
> In the leftmost subplot, we observe that the same observational input often corresponds to multiple real global state clusters at distinct locations in the 2D plane, forming several prominent Gaussian modes. The latent variable $z$ can thus be interpreted as a positional encoding for the coordinates of these Gaussian modes in the embedded space. The three subplots on the right display the global states generated by three selected agents, which faithfully reconstruct the multi-modal characteristics of the true state distribution. Notably, the location, shape, and amplitude of the reconstructed Gaussian modes closely resemble those of the real state distribution, empirically demonstrating $z$'s role in capturing semantic variations.
>
> **W7: The computational overhead of running a multi-step denoising process during execution.**
>
> **A for W7:**
> Thank you for raising this important point regarding computational overhead. We acknowledge these challenges and have designed GlobeDiff with practical deployability in mind, while also proposing concrete improvements for real-time applications as follows:
>
> - Efficient Architecture: GlobeDiff employs a lightweight model architecture. The diffusion model uses a compact U-Net with only two residual blocks, and the prior/posterior networks are shallow 3-layer MLPs operating on a low-dimensional latent space ($z_{\rm dim}$=16).
> - Fast Inference: Our inference process is highly efficient, requiring only $K=5$ denoising steps—substantially fewer than the 50-100+ steps common in generative imaging. Our ablation study (Figure 10) confirms that $K=5$ strikes an optimal balance between accuracy and latency.
>
> The experimental results in Table 1 show that although GlobeDiff's inference time is slightly longer than vanilla MAPPO, it remains below 12 milliseconds per timestep across various challenging environments. For future work, we will consider adaptive inference step reduction to dynamically decrease denoising steps (e.g., from $K=5$ to $K=2$) when system latency thresholds are exceeded.
>
>
> | |MMM2|3s|6h|zerg 5v5|zerg 10v10|zerg 10v11|protoss 5v5|terran 5v5|terran 10v11|
> |:-:|:-:|:-:|:-:|:-:|:-:|:-:|:-:|:-:|:-:|
> |vanilla MAPPO|2.69|2.85|2.97|2.73|2.74|2.87|2.66|2.62|2.90|
> |GlobeDiff|10.61|10.85|11.07|9.89|11.52|11.94|9.51|9.77|11.43|
>
> Table 1. Comparison of the inference time per time-step for vanilla MAPPO and GlobeDiff across different maps (unit: milliseconds).

---

> ### Author Response · Authors · 2025-11-21
> **Response to Reviewer mKZR (III)**
>
> **W8: A discussion of the method's limitations.**
>
> **A for W8:**
> We agree that a discussion of limitations is crucial for a balanced perspective.  The limitations of GlobeDiff primarily lie in two aspects.  First, the iterative denoising inherent in GlobeDiff can become computationally demanding as state dimensions grow.  While our U-Net architecture is efficient for typical SMAC tasks, it may face challenges with raw visual inputs or simulations involving thousands of variables.  The linear scaling of computation and memory with denoising steps is a key bottleneck in such high-dimensional scenarios.  Future work will explore techniques like latent-space diffusion to improve scalability.  On the other hand, while GlobeDiff handles ambiguity better than unimodal methods, capturing a vast number of equiprobable global states or exceptionally complex one-to-many mappings from a single observation remains challenging.  Future research will investigate more expressive priors (e.g., mixture models) to better model extreme multi-modality.
>
> We have incorporated this discussion of limitations into the revised manuscript to provide a more balanced view and guide future research directions.
>
>
> **Q2: Have you tested scenarios with limited or noisy communication?**
>
> **A for Q2:**
> As suggested, we have conducted additional experiments to investigate the influence of noisy communication. Specifically, we introduced a standard Gaussian noise scaled to 20\% of the observed value to each agent's local observation during the online training process, which we refer to as GlobeDiff (noise). This simulates significant sensor inaccuracies and perceptual uncertainty common in real-world scenarios.
>
> The experimental results in Table 2 show that compared to the vanilla setting, GlobeDiff in the noisy setting exhibited only a marginal performance decline. The observed minor performance drop demonstrates that our method exhibits a notable degree of robustness to significant observational noise, which can be attributed to the inherent properties of the diffusion-based state inference framework. In summary, first, our approach encodes task-relevant information into $z$, which helps ensure that task-relevant content remains consistent between noisy and denoised communication. Second, the diffusion process is inherently robust because it is trained to reverse a noise-addition process, making it resilient to input perturbations.
>
> | | zerg 5v5 | zerg 10v10|
> |:-:|:-:|:-:|
> |GlobeDiff (vanilla) |0.33$\pm$0.01|0.25$\pm$0.01|
> |GlobeDiff (noise)|0.31$\pm$0.01|0.23$\pm$0.01|
>
> Table~2. Comparison results of GlobeDiff on the vanillar and noisy communication setting.
>
> **Q3: How often do you expect the mode separation condition to hold in practice, and what is the empirical performance of GlobeDiff when the modes of the global state distribution are less well-separated?**
>
> **A for Q3:**
> The mode separation condition $ D > 4\sqrt{C_1K\delta^2 + C_2\varepsilon_{\text{KL}} + \sigma_{\text{max}}^2} $ is a sufficient condition for guaranteeing the tight error bound in Theorem 2. When this condition is not satisfied (i.e., when modes are less separated or overlapping), the theoretical bound in Theorem 2 becomes looser due to the exponential term $ \mathcal{O}(e^{-D^2/(8\sigma_{\text{max}}^2)}) $ becoming non-negligible. The core reason lies in the fact that within each Voronoi region, the error in approximating $p_i(s)$ by a unimodal Gaussian distribution $\mathcal{N}(\mu_i(x), \Sigma_i(x))$  increases, as indicated by the formula:
>
> \begin{equation}
> W_2^2(p_i(s), \mathcal{N}(\mu_i(x), \Sigma_i(x))) \leq \mathcal{O}\left(e^{-D^2/(8\sigma_{\text{max}}^2)}\right).
> \end{equation}
>
> The above analysis indicates that when modes are less separated, the theoretical guarantee for multi-modal coverage becomes weaker, and the model may not perfectly isolate each mode in the generated samples. However, the distribution-level form given by Theorem~1 still holds. This means that the overall bias between the generated states and the true states remains constrained, allowing the model to still produce good generation results, albeit with potentially less precise mode isolation.
>
> Thank you again for your thorough and insightful feedback, which has been instrumental in improving our work. We hope our responses and the additional results have fully addressed your concerns.

---

> ### Author Response · Authors · 2025-11-27
> **Looking forward to further discussions!**
>
> Dear reviewer,
>
> We were wondering if our response and revision have cleared all your concerns. In the previous response, we have tried to address all the points you have raised. We would appreciate it if you could kindly let us know whether you have any other questions, so that we can still have time to respond and address. We are looking forward to discussions that can further improve our current manuscript. Thanks!
>
> Best regards,
>
> The Authors

---

### Official Review · Reviewer_m4w8 · 2025-11-02

**Soundness:** 3
**Presentation:** 3
**Contribution:** 3
**Rating:** 6
**Confidence:** 3

**Summary:**

The paper presents a novel method “GlobeDiff” for tackling partial observability in multi-agent systems by framing global state inference as a conditional, multi-modal denoising process. The proposed GlobeDiff algorithm leverages diffusion models to generate a belief over the global state from local observations, enhanced by a latent variable to handle ambiguity. The paper is supported by theoretical error bounds and extensive empirical evaluation on modified StarCraft Multi-Agent Challenge (SMAC) environments.

**Strengths:**

1. Novel and Innovative Technical Approach: The core idea of applying diffusion models to the problem of global state inference in Dec-POMDPs is highly innovative. Instead of traditional belief estimation or communication, the authors formulate the problem as a conditional generative task. The introduction of a latent variable z to model the one-to-many mapping from local observations x to potential global states s is a sophisticated and technically sound solution to a fundamental challenge in this domain.

2. Strong Theoretical Foundation: The paper provides rigorous theoretical analysis, which is often lacking in deep learning-based MARL research. Theorems 1 and 2 establish formal bounds on the estimation error for both unimodal and multi-modal distributions. This theoretical grounding significantly strengthens the claims of the method's efficacy and provides insight into the conditions under which it is expected to perform well.

3. Comprehensive and Convincing Experimental Evaluation: 1) The proactive modification of the SMAC benchmark to create a more challenging and appropriate testbed for partial observability (SMAC-v1/2 PO). 2) Extensive comparisons against a diverse set of strong baselines (LBS, Dynamic Belief, CommFormer) and ablations against other generative models (VAE, MLP).

**Weaknesses:**

1. Dependence on Offline Data and Potential Distribution Shift: The training mechanism relies on an initial offline dataset to pre-train the GlobeDiff model. In many practical MARL scenarios, such a representative offline dataset may not be available. Furthermore, while the model is updated online, the initial dependence on offline data and the potential for distribution shift between offline and online data phases remain non-trivial challenges that are not fully addressed.

2. High Computational Complexity and Practical Overhead: Training and maintaining three separate components—the diffusion denoising network ϵθ, the prior network p_ϕ, and the posterior network q_ψ—is computationally expensive. The multi-step iterative denoising process during inference also adds latency. This complexity could be a major barrier to real-world deployment, especially in applications requiring low-latency decision-making or with limited computational resources.

**Questions:**

1. how to apply the methods to a latency-aware environment?
2. The authors compare against generative model baselines like VAE and MLP. However, given that you are using a history of observations (Eq. 1) in one scenario, how would a powerful sequence model like a Transformer [1,2], trained to directly predict the global state from the historical trajectory, perform as a baseline?

[1] Sequential Asynchronous Action Coordination in Multi-Agent Systems: A Stackelberg Decision Transformer Approach
[2] PTDE: Personalized training with distilled execution for multi-agent reinforcement learning

---

> ### Author Response · Authors · 2025-11-21
> **Response to Reviewer m4w8 (I)**
>
> Dear Reviewer,
>
> Thank you for your thorough review and insightful comments. Your feedback has been invaluable in helping us strengthen the paper. We have conducted additional experiments and analysis to address your concerns, and we believe the following responses provide the necessary clarifications.
>
> **W1: Dependence on Offline Data and Potential Distribution Shift.**
>
> **A for W1:**
> We appreciate you raising this important point regarding data requirements and distribution shift. We address this concern from two perspectives: data acquisition and online adaptation.
>
> - Data Acquisition: Our method can only requires an initial dataset generated from random policy rollouts. No expert demonstrations or domain-specific data collection strategies are needed. This minimal prerequisite ensures broad applicability, as such data can be easily collected through brief, unstructured environmental interaction, even in settings without pre-existing historical data.
>
> - Online Adaptation: Furthermore, our framework is designed to actively mitigate distribution shift through online updates. The diffusion model is updated periodically (every 50 episodes) with newly collected trajectories, allowing it to dynamically adapt to the evolving state distribution encountered by the learning agent. As shown in Figure 5 of the main paper, the progressive alignment between the inferred and true global states throughout training confirms that our model effectively adapts to these shifts.
>
>
> **W2 and Q1: High Computational Complexity and Practical Overhead.**
>
> **A for W2 and Q1:**
> We appreciate you raising this important point. Computational efficiency was a key consideration in the design of GlobeDiff. We address this concern by highlighting several design choices that minimize overhead, and we provide detailed empirical measurements of both training and inference latency.
>
> - Efficient Architecture: GlobeDiff employs a lightweight model architecture. The diffusion model uses a compact U-Net with only two residual blocks, and the prior/posterior networks are shallow 3-layer MLPs operating on a low-dimensional latent space ($z_{\rm dim}$=16).
> - Fast Inference: Our inference process is highly efficient, requiring only $K=5$ denoising steps—substantially fewer than the 50-100+ steps common in generative imaging. Our ablation study (Figure 10) confirms that $K=5$ strikes an optimal balance between accuracy and latency.
> - Infrequent Updates: During online deployment, the diffusion model is updated only once every 50 episodes, amortizing the training cost over long periods of interaction and minimizing online training overhead.
>
> To quantify this, we benchmarked the computational cost. The results in Table 1 show that GlobeDiff adds a modest ~30\% overhead to the total training time compared to vanilla MAPPO. While GlobeDiff's inference latency is higher (Table 2), it remains well within practical limits for real-time control, at under 12 milliseconds per step. As a direction for future work, we are exploring techniques like adaptive inference to dynamically reduce denoising steps (e.g., from $K=5$ to $K=2$) when tighter latency constraints are required.
>
> | |MMM2|3s|6h|zerg 5v5|zerg 10v10|zerg 10v11|protoss 5v5|terran 5v5|terran 10v11|
> |:-:|:-:|:-:|:-:|:-:|:-:|:-:|:-:|:-:|:-:|
> |vanilla MAPPO|14.1|14.0|15.2|15.4|15.9|19.2|15.1|15.8|16.9|
> |GlobeDiff|18.5|19.1|20.5|22.1|22.6|25.2|22.7|22.5|24.1|
>
> Table 1. Comparison of total training time (in hours). GlobeDiff introduces a modest overhead.
>
> | |MMM2|3s|6h|zerg 5v5|zerg 10v10|zerg 10v11|protoss 5v5|terran 5v5|terran 10v11|
> |:-:|:-:|:-:|:-:|:-:|:-:|:-:|:-:|:-:|:-:|
> |vanilla MAPPO|2.69|2.85|2.97|2.73|2.74|2.87|2.66|2.62|2.90|
> |GlobeDiff|10.61|10.85|11.07|9.89|11.52|11.94|9.51|9.77|11.43|
>
> Table 2. Comparison of inference latency per time-step (in milliseconds). GlobeDiff remains practical for real-time use.

---

> ### Author Response · Authors · 2025-11-21
> **Response to Reviewer m4w8 (II)**
>
> **Q2: Comparison with powerful sequence model like a Transformer?**
>
> **A for Q2:**
> To address your excellent question, we conducted a new experiment comparing GlobeDiff with a state-of-the-art Transformer-based sequence model, STEER [1]. For a fair comparison, we integrated STEER into the same MAPPO framework and trained it under identical conditions.
>
> The results in Table 3 show that GlobeDiff consistently and significantly outperforms STEER across all scenarios. We attribute this superior performance to the inherent strengths of the diffusion process for this task: its stochastic, multi-step denoising is better suited for global state prediction under uncertainty than the deterministic, autoregressive generation of a Transformer. This finding highlights a key advantage of our chosen modeling approach.
>
>
> | | STEER | GlobeDiff |
> |:-:|:-:|:-:|
> |zerg 5v5|0.26$\pm$0.01|0.33$\pm$0.02
> |protoss 5v5|0.24$\pm$0.01|0.38$\pm$0.01
> |terran 5v5| 0.20$\pm$0.01|0.24$\pm$0.01
> |zerg 10v10| 0.15$\pm$0.02|0.25$\pm$0.01
> |zerg 10v11| 0.06$\pm$0.01|0.12$\pm$0.01
> |terran 10v11|  0.04$\pm$0.00|0.07$\pm$0.01
> |MMM2| 0.32$\pm$0.07|0.49$\pm$0.11
> |3s5z vs 3s6z| 0.21$\pm$0.01|0.28$\pm$0.04
> |6h vs 8z| 0.10$\pm$0.01|0.47$\pm$0.04
>
> Table 3. Performance comparison between STEER (Transformer) and GlobeDiff (Diffusion).
>
> Thank you again for your timely and valuable feedback. Your suggestions have been instrumental in improving the paper, and we hope our detailed responses and new experiments have fully addressed your concerns.
>
> **Reference**
>
> [1] Zhang, Bin, et al. "Sequential asynchronous action coordination in multi-agent systems: A stackelberg decision transformer approach." Forty-first International Conference on Machine Learning. 2024.

---

> ### Author Response · Authors · 2025-11-27
> **Looking forward to further discussions!**
>
> Dear reviewer,
>
> We were wondering if our response and revision have cleared all your concerns. In the previous response, we have tried to address all the points you have raised. We would appreciate it if you could kindly let us know whether you have any other questions, so that we can still have time to respond and address. We are looking forward to discussions that can further improve our current manuscript. Thanks!
>
> Best regards,
>
> The Authors

---

> > ### Comment · Reviewer_m4w8 · 2025-11-28
> > **the response addresses my concerns**
> >
> > I read the comments from other reviewers and the responses from the authors.
> >
> > The responses addressed my concerns. Specifically, the additional experiments results in table 1/2/3 show strong advantages  over other methods.
> >
> > I would like to raise my score for supporting the acceptance of this paper.
> >
> >
> > ps: the authors are encourgaged to include the results in table 1/2/3 in the final paper.

---

> > > ### Author Response · Authors · 2025-11-29
> > > **Thanks for raising the score!**
> > >
> > > Dear Reviewer
> > >
> > > We would like to thank the reviewer for raising the score! We sincerely appreciate your time, valuable comments, and thoughtful engagement throughout the review process, which helped us significantly improve the paper's strengths. We will incorporate all feedback in the final version. We are grateful for your support and wish you all the best.

---

### Meta-Review · Area_Chair_E4cu · 2026-01-07

**Summary:**

The reviewers’ main concerns have focused on novelty and positioning. They have argued that using diffusion models for multi-agent partial observability is not entirely new and must be clearly discussed in the context of the existing literature. The second criticism has been providing more insights on the theoretical guarantees, especially in the context of standard ELBO/variational formulations. Finally, several comments have been made about the fairness of the empirical comparisons provided. There are also additional comments about the clarity of the presentation, especially regarding the exposition.

In the rebuttals, the authors have provided more discussions on the relevance to the existing literature (especially diffusion-for-PO related work) and have modified the statements on the claimed contributions and novelty. Furthermore, they have  strengthening experiments by comparing against additional relevant basedlines. Notably, the authors have also improved the presentation.

**Reviewer Concerns:**

Almost all the reviewers' concerns have been recognized, discussed in detail, and acted on by providing additional experiments, modifying the contribution statements, and clarifying the exposition.

**Reviewer Scores:**

Two of the reviewers have responded and mentioned that they would increase their ratings. The third reviewer has not engaged any further beyond their initial review. However, judging from their initial set of comments and the responses provided by the authors, I believe the key concerns -- which were shared by other reviewers -- are addressed.

---

### Decision · Program_Chairs · 2026-01-26

Accept (Poster)